# Measuring and modeling investigation of the Net Photochemical Ozone Production Rate via an improved dual-channel reaction chamber technique

Yixin Hao[1,2#], Jun Zhou[1,2#*], Jie-Ping Zhou[1,2], Yan Wang[1,2], Suxia Yang[1,2], Yibo Huangfu[1,2], Xiao-Bing Li[1,2], Chunsheng Zhang[3], Aiming Liu[3], Yanfeng Wu[1,2], Yaqing Zhou, Shuchun Yang[1,2], Yuwen Peng[1,2], Jipeng Qi[1,2], Xianjun He[1,2], Xin Song[1,2], Yubin Chen[1,2], Bin Yuan[1,2*], Min Shao[1,2]

[1]Institute for Environmental and Climate Research, Jinan University, Guangzhou 511443, China

[2]Guangdong-Hongkong-Macau Joint Laboratory of Collaborative Innovation for Environmental Quality, Guangzhou 511443, China

[3]Shenzhen National Climate Observatory, Shenzhen 518040, China

[#]Jun Zhou and Yixin Hao contribute equally to this work.

*Correspondence to*: Jun Zhou (Junzhou@jnu.edu.cn) and Bin Yuan (byuan@jnu.edu.cn)

**Abstract**. Current process-based research mainly uses box models to evaluate the photochemical ozone production and destruction rates, and it is unclear to what extent the photochemical reaction mechanisms are elucidated. Here, we modified and improved a net photochemical ozone production rate (NPOPR, $P(O_3)_{net}$) detection system based on the current dual-channel reaction chamber technique, which makes the instrument applicable to different ambient environments, and its various operating indicators were characterized, i.e., airtightness, light transmittance, wall losses of the reaction and reference chambers, conversion rate of $O_3$ to $NO_2$, air residence time, and performance of the reaction and reference chambers. The limits of detection of the NPOPR detection system were determined to be 0.07, 1.4, and 2.3 ppbv h$^{-1}$ at sampling flow rates of 1.3, 3, and 5 L min$^{-1}$, respectively. We further applied the NPOPR detection system to field observations at an urban site in the Pearl River Delta (China). During the observation period, the maximum value of $P(O_3)_{net}$ was 34.1 ppbv h$^{-1}$, which was ~ 0 ppbv h$^{-1}$ at night within the system detection error and peaked at approximately noon local time. The daytime (from 6:00–18:00) average value of $P(O_3)_{net}$ was 12.8 ($\pm$5.5) ppbv h$^{-1}$. We investigated the detailed photochemical $O_3$ formation mechanism in the reaction and reference chambers of the NPOPR detection system using a zero-dimensional box model. We found that the photochemical reactions in the reaction chamber were very close to those in ambient air, but it was not zero chemistry in the reference chamber, the reaction related to the production and destruction of $RO_2$ (=$HO_2$+$RO_2$) continued in the reference chamber, which

led to a small amount of $P(O_3)_{net}$. Therefore, the $P(O_3)_{net}$ measured here can be regarded as the lower limit of the real $P(O_3)_{net}$ in the atmosphere; however, the measured $P(O_3)_{net}$ was still $\sim$ 7.5 to 9.3 ppbv h$^{-1}$ higher than the modeled $P(O_3)_{net}$ value depending on different modeling methods, which may be due to the inaccurate estimation of $HO_2/RO_2$ radicals in the modeling study. Short-lived intermediate measurements coupled with direct $P(O_3)_{net}$ measurements are needed in the future to better understand $O_3$ photochemistry. Our results show that the NPOPR detection system can achieve high temporal resolution and continuous field observations, which helps us to better understand photochemical $O_3$ formation and provides a key scientific basis for continuous improvement of air quality in China.

## 1 Introduction

Surface $O_3$ pollution has become a major challenge in air quality management in China (Shen et al., 2021). Elevated surface $O_3$ mixing ratios exert severe adverse effects on public health, such as respiratory diseases, and the estimated annual mortality attributable to surface $O_3$ exposure exceeds 150,000 deaths in China (Malley et al., 2017). $O_3$ pollution is also detrimental to key staple crop yields, reducing the yields of wheat, soybean, and maize by up to 15 %, and is threatening global food security (Avnery et al., 2011; Mills et al., 2018; Karakatsani et al., 2010; Berman et al., 2012; O'Neill et al., 2003). As a greenhouse gas, $O_3$ also contributes significantly to climate change (Bell et al., 2004). With the rapid economic development and urbanization in the Pearl River Delta (PRD) region in China, $O_3$ pollution is pretty severe, especially in summer and autumn (Zou et al., 2015; Zhang et al., 2021).

The variation in $O_3$ in the planetary boundary layer is predominantly influenced by deposition, advection transport, vertical mixing (i.e., entrainment from the stratosphere), meteorological factors, and chemical reactions. Therefore, the $O_3$ budget in the boundary layer can be expressed as Eq. (1):

$$\frac{\partial [O_3]}{\partial t} = \underbrace{P(O_3) - D(O_3)}_{P(O_3)_{net}} \underbrace{- \frac{v}{H}[O_3]}_{SD} + \underbrace{u_i \frac{\partial [O_3]}{\partial x_i}}_{A} + STE \tag{1}$$

where SD, A, and STE represent the surface deposition, advection, and stratosphere-troposphere exchange (STE), respectively; $[O_3]$, $P(O_3)$ and $D(O_3)$ are the ambient $O_3$ mixing ratios, photochemical $O_3$ production and its loss rate, respectively; $v$, $H$, and $u_i$ represent the $O_3$ deposition velocity, mixing layer height, and velocity in three directions, respectively; and A consists of $u_i$ times the $O_3$ gradient in those three directions.

Tropospheric $O_3$ is a key component of photochemical smog, mainly formed by photochemical reactions of nitrogen oxides ($NO_x = NO + NO_2$) and volatile organic compounds (VOCs) (Lee et al., 2010). The specific process of the photochemical reaction is the photolysis of $NO_2$ at < 420 nm to generate $O(^3P)$ atoms, thereby promoting the formation of $O_3$ (Sadanaga et al., 2017). Simultaneously, there is a $RO_X$ ($RO_X = OH + HO_2 + RO_2$) radical cycle in the troposphere, which continuously provides $HO_2$ and $RO_2$ to oxidize NO to $NO_2$ resulting in the accumulation of $O_3$ (Shen et al., 2021; Sadanaga et al., 2017; Cazorla et al., 2010).

Typical meteorological scenarios for the occurrence of $O_3$ pollution episodes in polluted urban

centers are usually characterized by weak winds, strong solar radiation, and high temperature ($T$). Under
such conditions, local formation of $O_3$ plays a crucial role in the rapid increase of surface $O_3$ in daytime.
In addition, in Eq. (1), the surface deposition and advection of $O_3$ are proportional to ambient $O_3$ mixing
ratios, [$O_3$], which is mainly generated by local photochemistry (Carzorla et al., 2010). If $P(O_3)_{net}$ can be
reduced by regulatory measures, overall $O_3$ levels will decline proportionately over time (Cazorla et al.,
2010), thus, the investigation of $P(O_3)_{net}$ formation mechanism is urgently needed.

Current studies on $P(O_3)_{net}$ estimation mainly rely on modeling methods, the gas-phase chemical

mechanisms were frequently used to identify key drivers of $O_3$ pollution events and provide guidance for
making effective $O_3$ reduction strategies, such as the Master Chemical Mechanism (MCM), the regional
atmospheric chemistry mechanism (RACM), the Carbon Bond mechanisms (CBM) and the Mainz
Organic Mechanism (MOM) (Shen et al., 2021; Kanaya et al., 2016; Wang et al., 2014;  Tadic et al.,
2020; Ren et al., 2013; Lu et al., 2010; Zhou et al., 2014; Mazzuca et al., 2016). However, uncertainties
in emission inventories, chemical mechanisms, and meteorology make it difficult to perfectly reproduce
real atmospheric processes, which can lead some bias in modeling the $P(O_3)_{net}$. According to the existing
field observations, researchers found that the mixing ratios of $HO_2$ or $RO_2$ obtained from the model
simulation was inconsistent with that obtained from the direct measurement, leading to the deviation of
$P(O_3)_{net}$ between observation and model simulation results (Wang et al., 2014; Tadic et al., 2020; Ren et
al., 2013; Martinez et al., 2003). Therefore, we urgently need a method that can directly measure the
$P(O_3)_{net}$.

Recently, researchers have developed sensors that can directly measure $P(O_3)_{net}$ in the atmosphere

using the dual-channel chamber technique (Sadanaga et al., 2017; Cazorla et al., 2010; Baier et al., 2015
and 2017; Sklaveniti et al., 2018), where ambient air is introduced into two chambers of identical size,
one UV transparent chamber (reaction chamber) and one UV protection chamber (reference chamber).
In the presence of solar UV light, $O_3$ is produced by photochemical reactions in the reaction chamber,
but not in the reference chamber. The system does not directly measure $O_3$ mixing ratios, it measures the
combined mixing ratios of $O_3$ and nitrogen dioxide ($NO_2$). $P(O_3)_{net}$ is determined by the difference of
$O_X$($O_X=O_3+NO_2$) mixing ratios between the reaction and reference chambers. These studies have greatly
helped us to understand the $O_3$ photochemical formation mechanism, but defects still exists in current
studies, for example, the sensors developed by Cazorla et al. (2010) and Baier et al. (2015) both have an
$NO_2$-to-$O_3$ converter unit, and uses a modified $O_3$ monitor (Thermo Scientific, Model 49i, USA) to
measure Ox, but the zero point of the $O_3$ monitor is easy to drift, together with the limitation of the
conversion efficiency of $NO_2$ to $O_3$ (~ 99.9 %) and the effects of the $T$ and relative humidity (RH) to $O_3$
monitor, this method can introduce large measurement uncertainties. Sklaveniti et al. (2018) have shorten
the average residence time in the chambers to 4.5 min, which reduced the scattering and increased the
time resolution of $\Delta$Ox measurement, but large wall loss still exists in their system, which are 5 % and
3 % for $O_3$ and $NO_2$, respectively. Sadanaga et al. (2017) passed the NO standard gas into the PFA tube
to convert $O_3$ into $NO_2$ to detect Ox, which is easy to operate, but the LIF-$NO_2$ detector is less portable
and maintainable. Furthermore, all the current sensors have different degrees of wall loss of $NO_2$ and $O_3$
that can even reach 15 %, which largely affect the accuracy of the evaluation of $P(O_3)_{net}$.
In this study, we modified and improved a $P(O_3)_{net}$ sensor based on the dual-channel technique as
described above and named it the net photochemical ozone production rate (NPOPR) detection system.
Section 2 provides the improvement and characterization of the NPOPR detection system. Furthermore,
we applied the NPOPR detection system to an observation campaign conducted at Shenzhen
Meteorological Gradient Tower (SZMGT) in the Pearl River Delta (PRD) region in China. A zero-
dimensional box model based on the Framework for 0-D Atmospheric Modeling (F0AM) v3.2 coupled
with MCM v3.3.1 was used to simulate the photochemical reactions inside both the reaction and
reference chambers in the NPOPR system, which allowed us to assess the ability of the current modeling
method to model $P(O_3)_{net}$, as described in Sect. 3. The current research could help us study the source
and formation mechanism of $O_3$ and provide effective theoretical support for the prevention and control
of $O_3$ pollution. Because the system can directly obtain real-time $P(O_3)_{net}$ under different environmental
conditions, it can meet richer and more specific research needs.
**2 Method and materials**
**2.1 Development of the NPOPR detection system**
A schematic and actual diagram of the NPOPR detection system are shown in Fig. 1. The integral
construction is similar to the $P$-$L$(Ox) measurement system built by Sadanaga et al. (2017) and Sklaveniti
et al. (2018), which mainly consists of reaction and reference chambers with the same geometry and
made of quartz (190.5 mm inner diameter and 700 mm length; more details can be found in Fig. S1). To
prevent photochemical reactions inside the reference chamber, an ultraviolet (UV) protection Ultem film
(SH2CLAR, 3 M, Japan) was used to cover the outer surface to block sunlight with wavelengths < 390
nm. During the experiment, both the reaction and reference chambers were located outdoors and exposed
to sunlight directly to simulate genuine ambient photochemistry reactions. Ambient air was introduced
into the reaction and reference chambers at the same flow rate, and a Teflon filter was mounted before
the chamber inlet to remove fine particles. A stream of air from the two chambers was alternately
introduced into an NO-reaction chamber every 2 min to convert $O_3$ in the air to $NO_2$ in the presence of
high mixing ratios of NO ($O_3+NO=NO_2$), and the Ox mixing ratios from the outlet NO-reaction chamber,
i.e., the total $NO_2$ mixing ratios including the inherent $NO_2$ in the ambient and that converted from $O_3$,
were measured by a Cavity Attenuated Phase Shift (CAPS) $NO_2$ Monitor (Aerodyne research, Inc.,
Billerica MA, USA) to avoid other nitrogen oxide interferences to the $NO_2$ measurement (such as alkyl
nitrates, peroxyacyl nitrates, peroxynitric acid, nitrogen pentoxide, etc.). Compared to previous studies
that used a dual-channel UV-absorption $O_3$ monitor (Cazorla et al., 2010) or a laser-induced fluorescence
(LIF) LIF-$NO_2$ monitor (Sadanaga et al., 2017) for Ox measurements, our choice could make the NPOPR
detection system have a more stable zero-baseline and be more portable by assembling each part together,
i.e., put the CAPS $NO_2$ monitor, the automatic sampling system, and the automatic data sampling system
onto the indoor cabinets with the push-pull base, and put the dual chambers onto the outdoor shelf with
the push-pull base. Additionally, we modified the air sampling system to adjust the total air flow rates
freely from 1.3 to 5 L $min^{-1}$ in the reaction and reference chambers, which enabled us to achieve different
air residence times from 3.8 to 21 min. This time range covered all the residence times from previous
studies using different Ox measurement techniques, which ranged from 4.5 to 20.5 min (Cazorla et al.,
2010; Baier et al., 2015; Sadanaga et al., 2017; Sklaveniti et al., 2018). According to the simulation
results described in Sect. 3, the reaction rates of $O_3$ formation and destruction pathways and the radicals
that play critical roles in photochemical $O_3$ formation, such as $HO_2$, $RO_2$ and OH, reached quasi-steady
states in approximately 3 min, so it was reasonable for us to set the air flow rate highest at 5 L $min^{-1}$,
where the sampled air has already reacted for 3.8 min in the reaction and reference chambers. On the
other hand, this also demonstrated that it was reasonable to set the alternate sampling time for the reaction
and reference chambers at 2 min, where the sampled air actually has already reacted for at least 3.8 min
in the reaction and reference chambers. The switch system was controlled by two Teflon three-way
solenoid valves (001-0028-900, Parker, GER) located before the NO-reaction chamber (see Fig. 1). We
used homemade circuit control software (Four-Channel-Valves boxed) and a solenoid valve (001-0028-
900, Parker, GER) to automatically switch the sampling lines every 2 min. To keep the flow rates in the
reaction and reference chambers the same and avoid gas flow accumulation in the chamber, a pump
(pump 3) was connected to the Teflon three-way solenoid valves in parallel to the NO-reaction chamber
to evacuate the air that was not introduced into the NO-reaction chamber. To reduce NO interference, the
system used $O_X$ to infer the amount of $O_3$ generated by photochemical reactions (Liu et al., 1977; Pan et
al., 2015; Lu et al., 2010). The difference between the Ox mixing ratios in the reaction and reference
chambers, denoted by $\Delta Ox$, represents the amount of $O_3$ generated by the photochemical reaction.
$P(O_3)_{net}$ was obtained by dividing $\Delta Ox$ by the average residence time of air in the reaction chamber $\langle \tau \rangle$:
$$P(O_3)_{net} = P(O_X) = \frac{\Delta O_X}{\tau} = \frac{[O_X]_{reaction} - [O_X]_{reference}}{\tau} \qquad (2)$$
Igor Pro version 6 was used to calculate $P(O_3)_{net}$ as follows: ① separate the data of the reaction
and the reference chambers into two sets using the recorded valve number of 1 (reaction chamber) and 0
(reference chamber) during the sampling time; ② for each 2 min period of data, delete the first 20 s and
the last 20 s when the signal was not stable, then average the rest data, and do the interpolate calculation
of the reference chamber dataset; ③ calculate the difference between the Ox mixing ratios in the reaction
and reference chambers (i.e., $\Delta O_X$) at the time when the reaction chamber measured Ox; ④ divide $\Delta O_X$
by the average residence time of air in the reaction chamber $\langle \tau \rangle$ and obtain $P(O_3)_{net}$ at a time resolution
of 4 min.

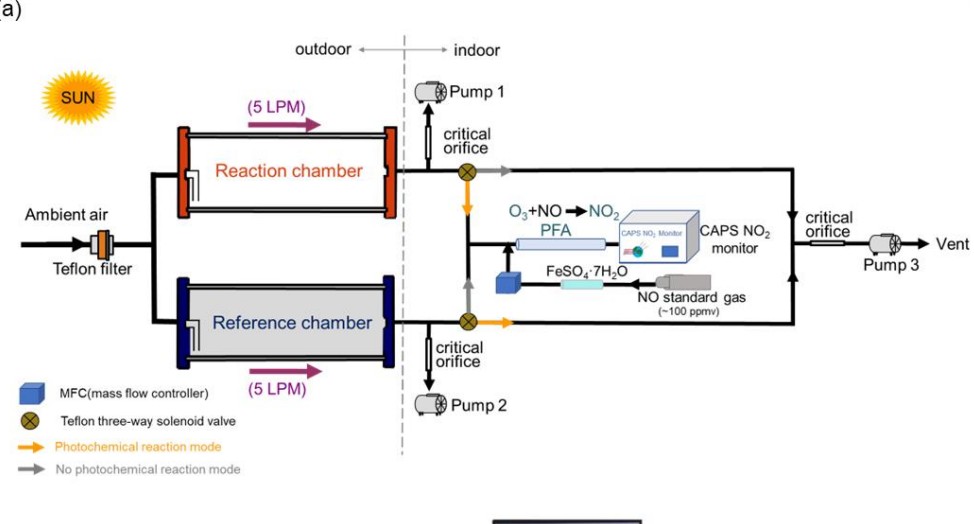

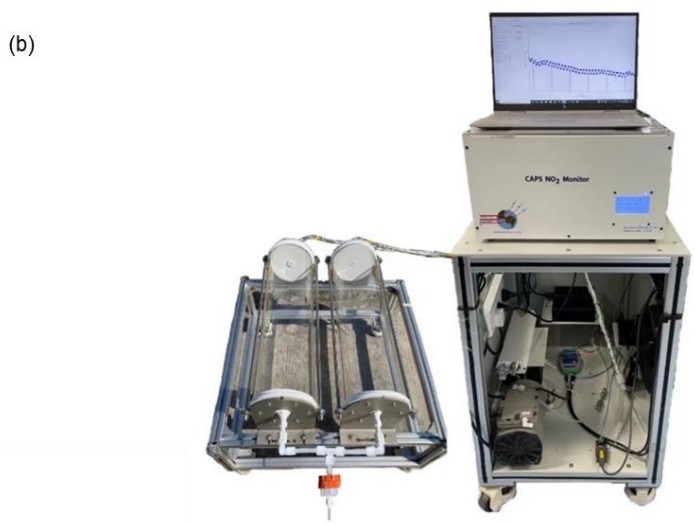


**Figure 1: (a) Schematic and (b) actual diagram of the NPOPR detection system.**

The major improvements of the NPOPR detection system described here compared to previous

studies to optimize $P(O_3)_{net}$ measurements are as follows: (1) we improved the design of the reaction and

reference chambers to ensure that they have good airtightness, which will increase the measurement

accuracy of different species inside the chambers. More details can be found in Sect. 2.2 and Appendix

I; (2) two pumps (labeled pump 1 and pump 2 in Fig. 1) were added directly after the reaction and

reference chambers to continuously draw ambient air through the two chambers (as the makeup flow) to

adjust the total air flow rates freely from 1.3 to 5 L min$^{-1}$ in the chambers. By doing this, we were able

to achieve different limits of detection (LODs) of the NPOPR system (see Sect. 2.4), making the

instrument applicable to different ambient environments, i.e., in highly polluted areas, we could use

higher air flow rates to reduce the wall loss effects of the chambers, and in less polluted areas, we could

use lower flow rates to increase the instrument LOD. (3) We characterized the NPOPR detection system

at different air flow rates (including 1.3, 2, 3, 4, and 5 L min$^{-1}$) and tested the conversion efficiency of $O_3$
by NO to $NO_2$ in the NO-reaction chamber at different NO mixing ratios and NO-reaction chamber
lengths. These efforts enabled us to better understand the running parameters of the NPOPR system and
perform data corrections under different air flow rates (see Sect. 2.2); (4) we tested the performance of
both the reaction and reference chambers by combining the field measurements and MCM modeling,
which indicated that reaction pathways of $P(O_3)$ and $D(O_3)$ and the radicals that play critical roles in
photochemical $O_3$ formation, such as $HO_2$, $RO_2$ and OH, reached quasi-steady states in approximately 3
min, thus ensuring that the lowest air residence time of 3.8 min (at an air flow rate of 5 L min$^{-1}$) in the
reaction and reference chambers was long enough to obtain accurate $P(O_3)_{net}$ values (see Sect. 3.2). These
efforts made the NPOPR system less prone to biases than other systems and increased its applicability.
**2.2 Characterization of the NPOPR detection system**
We characterized the NPOPR detection system following the same procedures as previous researchers,
including the residence time of the air, the wall losses of $NO_2$ and $O_3$, the transmittance of light and
temperature differences in the reaction and reference chambers, and the quantitative conversion
efficiency of $O_3$ to $NO_2$ ($\alpha$) in the NO-reaction chamber. Additionally, we investigated the residence time
of the air and the wall losses of $NO_2$ and $O_3$ in the reaction and reference chambers at different flow rates
(including 1.3, 2, 3, 4, and 5 L min$^{-1}$) and the conversion efficiency of $O_3$ by NO to $NO_2$ in the NO-
reaction chamber at different NO mixing ratios and NO-reaction chamber lengths. The detailed
experimental performances and data analysis are shown in Appendix I, and the corresponding results are
described as follows.
***The residence time.*** We tested the residence time of air in both chambers under different air flow
rates, including 1.3, 2, 3, 4, and 5 L min$^{-1}$, the obtained related residence time in the reaction chamber
were 0.35, 0.16, 0.11, 0.07, and 0.06 h, respectively. By setting different air flow rates, we were able to
obtain different residence time thus different limit of detection of NPOPR system, which make it
applicable to different ambient environment. To make sure that the mean residence time of air is the same
in the reaction and reference chambers, we also tested the residence time of air in the reference chamber
in parallel, which were not much difference with that in the reaction chamber, as shown in Table S1. The
experimental schematic diagram is shown in Fig. S2, the related results of different air flow rates are
shown in Fig. S3 and Table S1.

***Wall losses of NO₂ and O₃.*** At air flow rates of 1.3, 2, 3, 4, and 5 L min$^{-1}$, the wall losses of $O_3$ in

the reaction chamber were found to be approximately 2 %, 0 %, 0 %, 0 %, and 0 %, respectively, and the
wall losses of $O_3$ in the reference chamber were found to be approximately 2 %, 1 %, 1 %, 0 %, and
0.7 %, respectively. While the wall losses of $NO_2$ in the reaction chamber at air flow rates of 1.3, 2, 3, 4,
and 5 L min$^{-1}$ were found to be approximately 4 %, 4 %, 2 %, 0 %, and 0.3 %, respectively, the wall
losses of $NO_2$ in the reference chamber were found to be approximately 2 %, 1 %, 0 %, 0 %, and 0. 6 %,
respectively. The experimental schematic diagram is shown in Fig. S4, and the related results of different
air flow rates are shown in Figs. S5-S6 and Tables S2-S5. We further compared the wall losses of $O_3$ and
$NO_2$ in the reaction and reference chambers at 5 L min$^{-1}$ with previous studies (as shown in Table S6)
and found that they were significantly smaller, but even with a flow rate of 1.3 L min$^{-1}$, the wall losses
were still smaller than 4 % and 2 % in the reaction chamber and the reference chambers, respectively.
We also tested the wall losses of $NO_2$ and $O_3$ in the chamber at a 5 L min$^{-1}$ flow rate at different humidities
of 35-75 %, the detailed results are shown in Fig. S7 and S8, which shows that the variation in humidity
effected the wall loss of $NO_2$ and $O_3$ by 0.03-0.12 % and 1.06-1.19 %, respectively, which is much
smaller than the instrument detection error (which is 2 % at ambient $NO_2$ mixing ratios of 0-100 ppb),
which indicates the small effects of Ox loss on $P(O_3)_{net}$ measurements in our NPOPR detection system.

***The light transmittance and temperature differences in the reaction and reference chambers.*** It

is worth noting that there was still low transmittance of light ranging from 390 nm to 790 nm through
the UV protection film, and the reference chamber could not be regarded as completely dark; thus, we
tested the solar UV transmittance through the reaction and reference chambers of the NPOPR detection
system in the laboratory using a sunlight simulation lamp (SERIC XG-500B, Japan) to provide different
intensities of illumination. The photolysis frequencies of $NO_2$, $O_3$, HONO, $H_2O_2$, NO₃_M (photolysis of
$NO_3$ generates $NO+O_2$), NO₃_R (photolysis of $NO_3$ generates $NO_2+O$), HCHO_M (photolysis of HCHO
generates $H_2+CO$), and HCHO_R (photolysis of HCHO generates H+HCO) inside and outside the
reaction and reference chambers were measured using an actinic flux spectrometer (PFS-100; Focused
Photonics Inc., China). Table 1 presents the $J(NO_2)$, $J(O^1D)$, and $J(HONO)$ results for the outside and
inside chambers from this study and from the literature. $J(H_2O_2)$, $J(NO_3\_M)$, $J(NO_3\_R)$, $J(HCHO\_M)$,
and $J(HCHO\_R)$ are shown in Table S7.
The photolysis frequencies of all species inside the reaction chamber were in agreement with those
measured outside the reaction chamber within 4 %. Table S7 shows that the transmittivities of $J(H_2O_2)$,
$J(NO_3\_M)$, $J(NO_3\_R)$, $J(HCHO\_M)$, and $J(HCHO\_R)$ in the reaction chamber were more than 90 %.
However, we have observed that the transmittivities of $J(O^1D)$ were even higher than those of $J(HONO)$
(as shown in Table 1) in the reference chamber (which blocks sunlight at wavelengths < 390 nm),
theoretically, this is not possible according to JPL Publication 19-5 (Burkholder et al., 2020), where the
absorption cross section of HONO at wavelengths of 390-395 ranged from approximately $4.0\text{-}17.1 \times 10^{-21}$
$cm^2$, which is about two or three orders of magnitude higher than that of ozone (which ranged from
approximately $0.8\text{-}2.6 \times 10^{-23}$ $cm^2$ at wavelengths of 390-410 nm), and the photolysis quantum yield of
HONO at wavelengths of 390-395 is unity, which is about ten times higher than that of ozone (~ 0.08).
This will surely make the $J$ values of HONO inside the reference chamber (which only has sunlight with
wavelengths > 390 nm) higher than that of ozone, according to the Eq. (S9). We also found that the
transmittivity of HONO and $O_3$ in the reference chamber obtained from the TUV simulation (as described
in Sect. 3.2) were 0.01 and 0, respectively, as shown in Table S13. Therefore, we believe the non-zero
measurement results of the transmittivity of $O_3$ shown in Table 1 and Table S7 are mostly probably due
to the instrument measurement error, this error is relatively large due to a limit number of measurement
points (3 points for each species). We further evaluated the measurement error of $J$ values based on the
instrument measurement error of the actinic flux spectrometer, which can reach $\pm 5$ % according to Bohn
et al. (2017), and re-evaluated the transmittivity error listed in Table 1 and Table S7 following the
procedures described in supplementary materials (Sect. 1.5). The calculation result from Eq. (S5) show
that the transmittivities errors are 0.07 for all species, within this error range, $J(O1D)$, $J(HONO)$, $J(H_2O_2)$,
$J(HCHO\_M)$, and $J(HCHO\_R)$ can be considered statistically indistinguishable from 0 in reference
chamber. However, $J(NO_2)$, $J(NO_3\_M)$, and $J(NO_3\_R)$ still distinctly positive values. Specifically, the
transmittivities of $J(NO_3\_M)$ and $J(NO_3\_R)$ of the reference chamber were more than 90 % (Table S7).
The influence of the measurement error of $J$ values of all species on $P(O_3)_{net}$ will be discussed in Sect. 3.

**Table 1. Transmittivities of photolysis frequency $J$ (s$^{-1}$) values of different species in the reaction and reference chambers. The shaded and clear regions correspond to the transmittivities of $J$ values in the reference (Ultem coated) and reaction (clear) chambers, respectively. The "transmittivities" column shows the transmittivities of the tested species from the measurements conducted with the set photolysis frequencies using SERIC XG-500B sunlight (this study) and ambient (literature). It should be noted that the errors listed here are relatively large and may not reliable due to a limit number of measurement points (3 points for each species). The calculated transmittivity errors are 0.07 for all species based on the ±5 % measurement error of the instrument.**

| | Transmittivities | | | |
| --- | --- | --- | --- | --- |
| | Averaged (this study) | Cazorla *et al.*, 2010 | Baier *et al.*, 2015 | Sadanaga *et al.*, 2017 |
| $J(NO_2)$ | 0.985 ±0.037 | 0.974 | 0.990 | 0.986 |
| | 0.094 ±0.014 | 0.021 | 0.01 | 0.121 |
| $J(O^1D)$ | 1.020 ±0.04 | 0.991 | 0.978 | 1.030 |
| | 0.019 ±0.011 | 0.0058 | 0.001 | ~0 |
| $J(HONO)$ | 0.983 ±0.037 | 0.976 | 0.982 | 0.988 |
| | 0.002 ±0.0002 | 0.0067 | ~0 | 0.017 |

We further detected the temperature in both the reaction and reference chambers when running the NPOPR system in an ambient observation campaign during November 2022 on the Panyu campus of Jinan University in Guangzhou, China (113° 36′ E, 23° 02′ N). We found that the UV protection Ultem film on the reference chamber did not block the heat outside the chamber, and the temperature remained the same in the reaction and reference chambers during the measurement test, as shown in Fig. S10.

***The quantitative conversion efficiency of O₃ to NO₂ (α)*** in the NO-reaction chamber is crucial for accurate measurement of $P(O_3)_{net}$. Here, we used a perfluoroalkoxy (PFA) tube (outer diameter of 12.7 mm; inner diameter of 9.5 mm) as the NO-reaction chamber. The experimental schematic diagram is shown in Fig. S12. Known mixing ratios of $O_3$ and NO standard gas were introduced into the NO-reaction chamber, and NO reacted with $O_3$ to produce $NO_2$. To avoid the influence of small amounts of $NO_2$ impurity in the NO standard gas used for conversion, we added a cylinder filled with partialized crystals of $FeSO_4 \cdot 7H_2O$ to reduce $NO_2$ in the NO/$N_2$ gas cylinder to NO. We injected ~1800 ppbv NO into the NO-reaction chamber and tested the $NO_2$ mixing ratios from its outlet using a CAPS $NO_2$ monitor, as shown in Fig. S13. We found that the standard deviation of the $NO_2$ mixing ratios was lower than 0.027

ppbv, which is smaller than the baseline drifts of the CAPS (which were 0.043 and 0.030 ppbv (1 $\sigma$) at
integration times of 35 and 100 s, respectively, as mentioned in Sect. 2.3), so we believe the particulate
crystals of $FeSO_4 \cdot 7H_2O$ performed well and the potential bias introduced by the impurity in NO mixing
ratio for $P(O_3)_{net}$ was negligible. Finally, the total $NO_2$ mixing ratios, including that from the ambient air,
were measured using a CAPS $NO_2$ monitor ($[NO_2]_{CAPS}$). The $O_3$ mixing ratios were controlled at
approximately 310 ppbv according to the maximum mixing ratio range in the normal ambient atmosphere
(to ensure that all ambient and newly generated $O_3$ can react with NO and produce equivalent amounts
of $NO_2$). An $O_3$ generator equipped with a low-pressure mercury lamp was employed to generate $O_3$, and
the generated $O_3$ mixing ratios ($[O_3]_g$) were measured by a 2B $O_3$ monitor as mentioned above. Here, we
note that the $O_3$ mixing ratios were diluted by the added $NO/N_2$ gas (with a flow rate of 20 mL min$^{-1}$) in
the NO-reaction chamber (with a total flow rate of 1.11 L min$^{-1}$), taking 1800 ppbv $NO/N_2$ gas as an
example, the relationship between $[NO_2]_{CAPS}$ and $[O_3]_g$ can be described by Eq. (3):
$$[NO_2]_{CAPS} = \frac{1.09}{1.11}[O_3]_g \alpha \qquad (3)$$
To determine the optimal length of the NO-reaction chamber and NO mixing ratios, we performed
a cross test of $\alpha$ under the following scenarios: the NO-reaction chamber lengths were increased from 30
to 650 cm in 50 cm steps, and the NO standard gas (102.1 ppmv) was diluted to 600, 900, 1200, 1500,
1800, 2100, and 2400 ppbv in the NO reaction chamber. The results are shown in Fig. 2. We found that
at $[O_3]_g$ of approximately 310 ppbv, with NO mixing ratios $\geq$1800 ppbv in the NO reaction chamber, $\alpha$
reached 99 %, 99.6 %, and 99.9 % with NO-reaction chamber lengths of 50, 70, and 100 cm, respectively,
where the corresponding $O_3$ residence times in the NO reaction chamber were 1.95, 2.74, and 3.91 s,
respectively. Considering both the optimal reaction time in the NO reaction chamber and $\alpha$, we selected
the NO reaction chamber length as 100 cm with an NO mixing ratio of 1800 ppbv for the NPOPR
detection system.

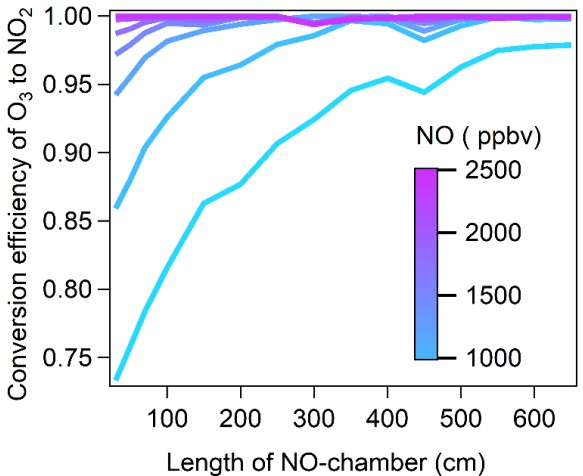

**Figure 2: The conversion efficiency of O₃ by NO to NO₂ in the NO reaction chamber as a function of the NO chamber length, color coded by the NO mixing ratios.**

***The airtightness of the reaction and reference chambers*** We also checked the airtightness of the reaction and reference chambers by passing through gases with different flow rates based on the schematic diagram shown in Fig. S14 and compared the values of [air flow rate × pressure] between the inlet and outlet of the chambers (as indicated in Fig. 3). We found that the deviations in [air flow rate × pressure] at the inlet and outlet of the reaction and reference chambers at different flow rates were <3 % (as shown in Table S8), indicating the good airtightness of the reaction and reference chambers. This ensured that the photochemical reactions in the reaction and reference chambers would not be affected by the ambient air outside the chambers.

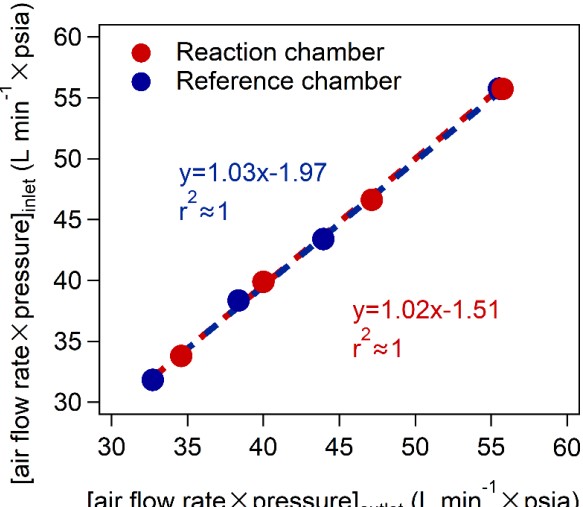

**Figure 3: The relationship of the values of [air flow rate × pressure] between the inlet and outlet of the chambers (psia: Pounds Per Square Inch Absolute).**

***The flow states in the reaction and reference chambers*** We calculated the Reynolds number to

check the gas flow state in the reaction and reference chambers. The Reynolds number (expressed as Eq.

(4)) is a dimensionless number that can be used as the basis for judging the flow characteristics of a fluid:

$$Re = \rho v d / \mu \tag{4}$$

where $v$, $\rho$ and $\mu$ are the flow velocity, density and viscosity coefficient of the fluid, respectively. In this

study, the fluid is air, and d is the equivalent diameter of the reaction and reference chambers. The

calculated Reynolds numbers in the two chambers under flow rates of 1.3, 2, 3, 4, and 5 L min$^{-1}$ were

9.39, 14.58, 21.75, 29.05, and 36.34, respectively, indicating laminar flows in both chambers at different

flow rates.

***The HONO production in the reaction and reference chambers*** We tested the HONO production

in the reaction and reference chambers under weather conditions similar to those during the SZMGT

observations (humidity of 60-90% at a temperature of ~ 20 °C and $J(NO_2)$ of ~ 0-8 × 10$^{-3}$ s$^{-1}$) at a 5 L

min$^{-1}$ sampling flow rate. We found that the HONO mixing ratios in the reaction and reference chambers

were almost the same and not statistically different from that in the ambient air within the standard

deviation, as shown in Fig. S9; therefore, we assumed that the HONO production in the reaction and

reference chambers would not cause a significant difference in $P(O_3)_{net}$ in the two chambers.

Unfortunately, we did not test HONO during the field observation period, but we have added the modeled

HONO produced from the precursors before the ambient air was injected into the NPOPR system, as

described in Sect. 3.2.

## 2.3 Calibration and measurement error of the CAPS NO$_2$ monitor

The Ox in the NPOPR detection system was measured by the CAPS NO$_2$ monitor. Detailed descriptions

of this technique can be found elsewhere (Kebabian et al., 2008, 2005). We calibrated the CAPS NO$_2$

monitor using a NO$_2$ standard gas (with an original mixing ratio of 2.08 ppmv), which was first calibrated

using the gas-phase titration method using NO standard gas and excessive O$_3$. The related experimental

results are shown in Fig. S15. The detailed calibration procedure is as follows: a. injected ~ 10–100 ppbv

of NO$_2$ standard gas for 30 min to passivate the surfaces of the monitor and then injected dry pure air for

~ 10 min to minimize the zero point drift, which were 0.043 and 0.047 ppbv at integration times of 35

and 100 s, respectively, and resulted in LODs of CAPS of 0.13 and 0.14 ppbv (3 $\sigma$), respectively; b.

injected a wide range of $NO_2$ mixing ratios (from 0–160 ppbv) prepared by mixing the $NO_2$ standard gas
with ultrapure air into the CAPS $NO_2$ monitor and repeated the experiments three times at each $NO_2$
mixing ratio. The final results are shown in Fig. 4. To check the baseline drift of the CAPS at different
humidities, we added another two sets of tests (as shown in Fig. S11) using ambient air and wet pure air
and found that (a) when injecting ambient air into the CAPS (RH ranged from ~30-35%), the baseline
drifts were 0.035 and 0.032 ppbv (1 $\sigma$) at integration times of 35 and 100 s, respectively; and (b) when
injecting wet pure air into the CAPS (RH ranged from 35-70%), the baseline drifts were 0.043 and 0.030
ppbv (1 $\sigma$) at integration times of 35 and 100 s, respectively. These baseline drifts were smaller than those
when injecting dry pure air to estimate the LOD of the CAPS. We chose the largest baseline drift when
injecting dry pure air to estimate the $P(O_3)_{net}$ error in the following analysis; by doing this, we were able
to include all the short-duration baseline drifting in the CAPS $NO_2$ monitor under different humidities.
To obtain an accurate measurement error of the CAPS $NO_2$ monitor (($O_{X_{CAPS}})_{error}$), we fitted the
calibration results with a 68.3 % confidence level, and the blue line in Fig. 4 represents the maximum
fluctuation range under this confidence level. $(O_{X_{CAPS}})_{error}$ was then calculated from the fluctuation
range of the 68.3 % confidence interval of the calibration curve. The relationship between $(O_{X_{CAPS}})_{error}$
and the measured Ox value ([Ox]$_{measured}$) can be expressed as a power function curve, as shown in Eq.

(5):

$$(O_{X_{CAPS}})_{error} = 9.72 \times [O_X]_{measured}^{-1.0024} \tag{5}$$
Subsequent $P(O_3)_{net}$ error estimation according to the instrument measurement error of the CAPS
$NO_2$ monitor and the $O_3$ light-enhanced loss in the reaction and reference chambers are described in
Appendix II.

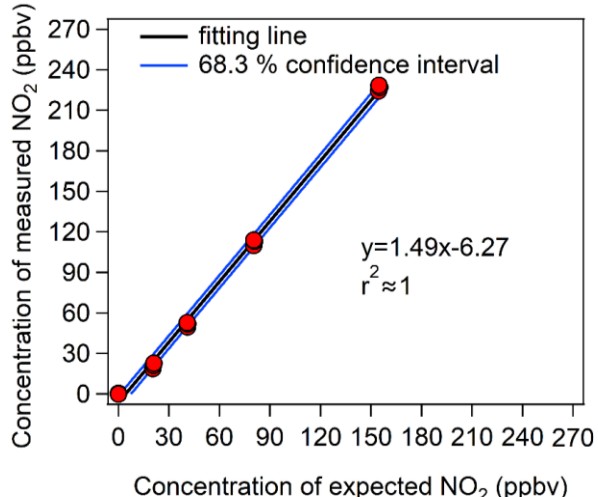

**Figure 4: Calibration results of the CAPS NO₂ monitor with different NO₂ mixing ratios. The y-axis represents the NO₂ mixing ratios measured by the CAPS NO₂ monitor, and the x-axis represents the prepared NO₂ mixing ratios from the diluted NO₂ standard gas.**

**2.4 The measurement error of $P(O_3)_{net}$ and the LOD of the NPOPR detection system**

To assess the measurement error of $P(O_3)_{net}$ and the LOD of the NPOPR detection system, dry pure air was introduced into the NPOPR detection system in sequence to adjust the system for ~ 2 h, followed by dry pure air or ambient air when the time resolution of the CAPS NO₂ monitor was 1 s and the integration time period was 100 s (the measurement durations for the reaction and reference chambers were both 2 min). The LOD of the NPOPR detection system was obtained as three times the measurement error of $P(O_3)_{net}$, which was determined at a time resolution of 4 min by propagating the errors of the Ox measured by the CAPS NO₂ monitor when ultrapure air or ambient air was introduced into the NPOPR detection system, combined with the related $\langle \tau \rangle$ under different flow rates, i.e., $\langle \tau \rangle$ was 0.063 h at a flow rate of 5 L min⁻¹. The detailed calculation method is shown in Eq. (6):

$$\text{LOD} = \frac{3 \times \sqrt{([O_X]_{rea\_std})^2 + ([O_X]_{ref\_std})^2}}{\tau} \tag{6}$$

where $[O_X]_{rea\_std}$ and $[O_X]_{ref\_std}$ represent the standard deviation of $O_X$ in the reaction and reference chambers measured by the CAPS NO₂ monitor with an integration time period of 100 s, respectively.

However, considering that the background Ox mixing ratios (measured by the CAPS NO₂ monitor of the air in the reference chamber) changed when measuring the ambient air, the measured $O_X$ errors in the reaction and reference chambers changed with the Ox mixing ratios (as shown in Sect. 2.3), and the LOD must also be a function of the intrinsic ambient and photochemically formed $O_3$ and NO₂ mixing

ratios (i.e., the Ox mixing ratios measured by the CAPS $NO_2$ monitor). It is worth noting that the
measured $O_X$ errors may also be influenced by the light-enhanced loss of $O_3$ in the reaction and reference
chambers under ambient conditions when the light intensity (especially $J(O^1D)$) and $O_3$ mixing ratios are
high, as tested and shown in Appendix II, but this effect is included in the measured $O_X$ errors. Therefore,
when injecting ambient air into the NPOPR system, the error and LOD of $P(O_3)_{net}$ with a residence time
of $\tau$ can be calculated using Eq. (7) and Eq. (8), respectively:
$$P(O_3)_{net\_error}=\frac{\sqrt{(O_{X_\gamma})_{rea\_error}^2+((9.72\times[(O_X]_{rea\_measured}^{-1.0024}))_{rea\_std}^2+(O_{X_\gamma})_{ref\_error}^2+((9.72\times[(O_X]_{ref\_measured}^{-1.0024}))_{ref\_std}^2}}{\tau} \tag{7}$$

$\text{LOD}= 3\times P(O_3)_{net\_error}$ (8)
where $(O_{X_\gamma})_{rea\_error}$ and $(O_{X_\gamma})_{ref\_error}$ represent the measurement error caused by the light-enhanced loss
of $O_3$ in the reaction and reference chambers, respectively, and $(9.72\times[O_X]_{measured}^{-1.0024})_{rea\_std}$ and
$(9.72\times[O_X]_{measured}^{-1.0024})_{ref\_std}$ represent the standard deviation of $O_X$ in the reaction and reference
chambers caused by the CAPS $NO_2$ monitor with an integration time period of 100 s, respectively. More
details about the $(O_{X_\gamma})_{rea\_error}$ and $(O_{X_\gamma})_{ref\_error}$ estimation method can be found in Appendix II.
In conclusion, the LOD of the NPOPR detection system is determined to be three times $P(O_3)_{net\_error}$,
where $P(O_3)_{net\_error}$ is mainly determined by the measurement error of Ox (including the Ox measurement
error of the CAPS $NO_2$ monitor, the light-enhanced loss of $O_3$, and the chamber Ox losses). Because the
measurement error of the CAPS $NO_2$ monitor decreases with increasing Ox mixing ratios (as shown in
Sect. 2.3), higher LODs could be obtained when injecting dry pure air into the NPOPR detection system,
which were approximately 0.07, 1.4, and 2.3 ppbv $h^{-1}$ at air flow rates of 1.3, 3, and 5 L $min^{-1}$, respectively.
The results are summarized in Table S9.
During the field observations, the LOD values were highly dependent on the ambient conditions,
especially the light intensity and the Ox mixing ratios, and higher $O_3$ mixing ratios and lower light
intensity will likely result in lower LOD values.
**2.5 Laboratory tests of the NPOPR detection system**
We conducted an experiment in the laboratory to test the performance of the NPOPR detection system at
Jinan University Panyu Campus (23.0° N, 113.4° E) on 26 March 2021. Ambient air (5 L $min^{-1}$) was
simultaneously injected into the reaction and reference chambers of the NPOPR detection system in
parallel, and the sunlight simulation lamp mentioned above was used to simulate sunlight radiation. The
light intensities of the sunlight simulation lamp were decreased from 26000 cd to 0 cd in steps of 3700
cd, where cd indicates the light intensity SI unit candela. $P(O_3)_{net}$ was 28.6 ppbv h$^{-1}$ at a light intensity of
26000 cd and gradually approached 0 ppbv h$^{-1}$ at 0 cd (as shown in Fig. 5), indicating that the $P(O_3)_{net}$
change due to the different sunlight radiation could be well captured by the NPOPR detection system.

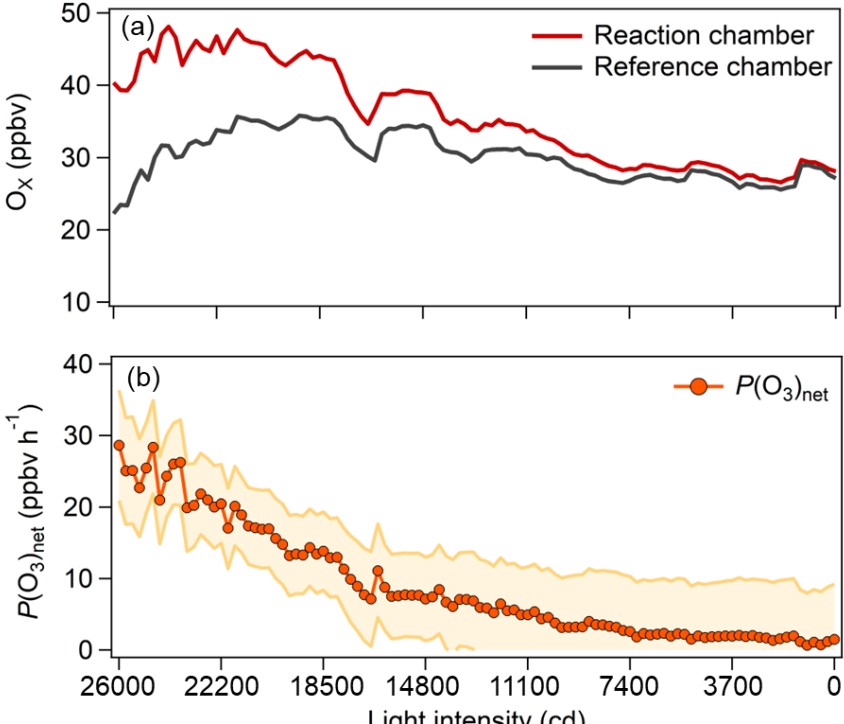


**Figure 5: (a) Measured Ox mixing ratios in the reaction and reference chambers and (b) the related $P(O_3)_{net}$**
**as a function of the light intensity during the experiment.**
**3 Atmospheric study and discussion**
**3.1 Field observations**
The self-built NPOPR detection system was employed in the field campaign conducted at SZMGT, which
is located in Shenzhen, China (as shown in Fig. S17), from 7 to 9 December 2021. During the campaign,
to achieve the lowest $O_3$ and $NO_2$ wall loss, we used a 5 L min$^{-1}$ air flow rate in the reaction and reference
chambers (with a residence time of ~ 4 min). The photolysis frequencies of different species were
measured using the actinic flux spectrometer as mentioned above. $O_3$ and $NO_X$ (NO+NO$_2$) mixing ratios
were measured using a 2B $O_3$ monitor and a chemiluminescence NOx monitor (Model 42i, Thermo
Fisher Scientific, USA), respectively. *T* and RH were measured by a portable weather station (Met Pak,
Gill Instruments Ltd, UK). Volatile organic compounds (VOCs) were measured by high-resolution
proton transfer reaction time-of-flight mass spectrometry (PTR-ToF-MS, Ionicon Analytik, Austria)
(Wang et al., 2020a; Wu et al., 2020) and an off-line gas chromatography mass spectrometry flame
ionization detector (GC–MS-FID) technique (Wuhan Tianhong, Co. Ltd, China) (Yuan et al., 2012)
(Table S11). Additionally, a self-built formaldehyde analyzer was used to detect formaldehyde (HCHO)
(Zhu et al., 2020). Figure 6 presents the temporal and diurnal variations in the $P(O_3)_{net}$, $O_X$, $O_3$, NO,
$NO_2$, $NO_X$, *T*, RH, $J(O^1D)$, and $J(NO_2)$ mixing ratios at SZMGT during the campaign.

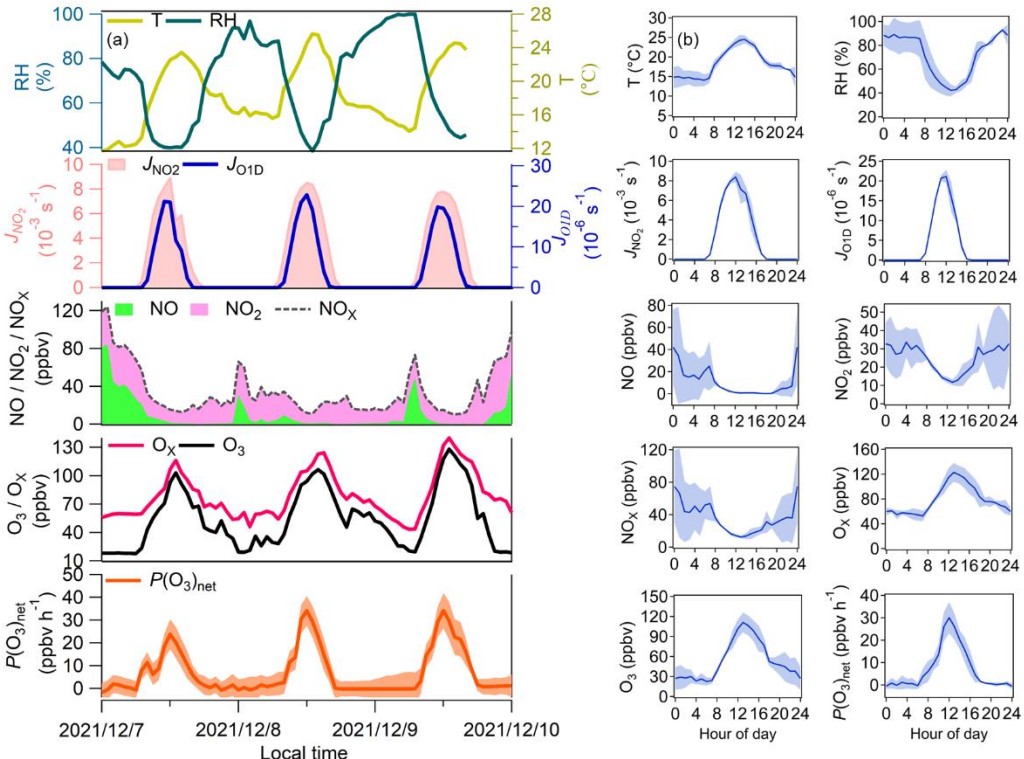


**Figure 6: (a) Time series and (b) average diurnal variations of $P(O_3)_{net}$, $J(NO_2)$, $J(O^1D)$, T, RH, $O_X$, $NO_2$ and**
**NO measured at SZMGT from 7 to 9 December 2021. The shaded areas represent the error of each measured**
**species, where the error of $P(O_3)_{net}$ was calculated according to the method described in Appendix II (the**
**estimation of the $P(O_3)_{net}$ error).**

During the measurement period, $P(O_3)_{net}$ ranged from ~ 0 to 34.1±7.8 ppbv h⁻¹, with an average
daytime (from 6:00–18:00) value of 12.8 (±5.5) ppbv h⁻¹. The maximum $P(O_3)_{net}$ at SZMGT was lower
than that measured in the urban area of Houston in the US (40–50 ppbv h⁻¹ and 100 ppbv h⁻¹ in autumn
and spring, respectively) (Baier et al., 2015; Ren et al., 2013), close to that measured in Indiana in the
US (~ 30 ppbv h⁻¹ in spring) (Sklaveniti et al., 2018), and much higher than that measured at the

473 Wakayama Forest Research Station, a remote area of Japan (10.5 ppbv h$^{-1}$ in summer) (Sadanaga et al.,

474 2017) and an urban area of Pennsylvania in the US (~ 8 ppbv h$^{-1}$ in summer) (Cazorla et al., 2010). The

475 result indicates the rationality of the measured $P(O_3)_{net}$ in this study. From previous studies, the $O_3$

476 pollution in the PRD area is more severe in summer and autumn than in winter and spring (Zhang et al.,

477 2021). In this study, $P(O_3)_{net}$ was measured in wintertime, which was already high, so we believe that the

478 $O_3$ pollution of the PRD is severe and urgently needs to be controlled. More measurements of $P(O_3)_{net}$

479 worldwide are listed in Table S10, and we found that $P(O_3)_{net}$ was much higher in urban areas than in

480 remote areas using both modeling and direct measurement methods.

481  According to the diurnal variation in all the measured pollutant indicators, $P(O_3)_{net}$ started to

482 increase at approximately 7:00 local time, which may be due to two reasons: (1) the rise in $O_3$ precursors

483 (i.e., VOCs) due to the high-altitude atmospheric residual layer transported downward near the surface

484 at this time and (2) the increase in solar radiation intensity after 7:00, which increased the oxidation

485 capacity of the pollutants. These two factors jointly enhanced the photochemical oxidation reaction of

486 VOCs and gradually increased $P(O_3)_{net}$. $P(O_3)_{net}$ then peaked at approximately 12:00, which was

487 consistent with $J(NO_2)$, but this peak time occured earlier than that of $O_3$, which peaked at approximately

488 14:00 , which may be due to the photochemical reactions dominating $O_3$ mixing ratio changes between

489 12:00 and 14:00 . After 14:00, the $O_3$ mixing ratios started to decrease, which may be due to other

490 processes dominating the $O_3$ mixing ratio changes at this time, such as $O_3$ reacting with other pollutants

491 or surface deposition and the outflow of $O_3$ by physical transport. In conclusion, the changes in $O_3$ mixing

492 ratios were influenced by both photochemical production and physical transport. Because $HO_2$ and $RO_2$

493 were not well captured in the model, the simulations could lead to an underestimation of $P(O_3)_{net}$.

494 **3.2 Model simulation of $P(O_3)_{net}$ in the reaction and reference chambers**

495 **3.2.1 Modeling method**

496 To obtain a comprehensive understanding of the ozone production rate $P(O_3)$ and ozone destruction rate

497 $D(O_3)$ during a 4-min photochemical reaction in the reaction and reference chambers, we modeled $P(O_3)$

498 and $D(O_3)$ at 12:00 on 7 December 2021 based on field observation data using a zero-dimensional box

499 model based on the F0AM v3.2 coupled with MCM v3.3.1, which contains a total of 143 VOCs, more

500 than 6700 species, involving more than 17000 reactions (Jenkin et al., 2015). $P(O_3)_{net}$ can be expressed

501 by the difference between $P(O_3)$ and $D(O_3)$, and $P(O_3)$ and $D(O_3)$ can be expressed as Eqs. (9) and (10),

respectively.
$P(O_3)=k_{HO_2+NO}[HO_2][NO]+\sum_i k_{RO_{2,i}+NO}[RO_{2i}][NO]\varphi_i$  (9)
$D(O_3)=k_{O(^1D)+H_2O}[O(^1D)][H_2O]+k_{OH+O_3}[OH][O_3]+k_{HO_2+O_3}[HO_2][O_3]$
$+\sum_i \left(k_{O_3+Alkene_i}[O_3][Alkene_i]+k_{OH+NO_2}[OH][NO_2]+k_{RO_{2,i}+NO_2}[RO_{2i}][NO_2]\right)$  (10)
where $k_{M+N}$ represents the bimolecular reaction rate constant of M and N, and $\varphi_i$ is the yield of $NO_2$ from
the $RO_{2i}$ +NO reaction. The relevant reaction rates of $P(O_3)$ and $D(O_3)$ and the VOCs mixing ratios from
7–9 December 2021 at the SZMGT used in the model are listed in Tables 2 and S11.

**Table 2. O₃ production and destruction reactions and the relevant reaction rates used in the model.**

| Reactions | Rate coefficient/unit | Number |
|---|---|---|
| O₃ production pathways - $P(O_3)$ | | |
| $RO_2+ NO\rightarrow RO + NO_2$ | $2.7\times10^{-12}\times exp(360/T)$/molecules $^{-1}$ cm$^3$ s$^{-1}$ | (R1) |
| $HO_2+ NO\rightarrow OH + NO_2$ | $3.45\times10^{-12}\times exp(270/T)$/molecules $^{-1}$ cm$^3$ s$^{-1}$ | (R2) |
| O₃ loss pathways - $D(O_3)$ | | |
| $O_3 + hv \rightarrow O^1D + O_2$ | Measured $JO^1D$/s$^{-1}$ | (R3) |
| $O_3 + C_2H_4 \rightarrow HCHO + CH_2OOA$ | $9.1\times10^{-15}\times exp(-2580/T)$/molecules $^{-1}$ cm$^3$ s$^{-1}$ | (R4) |
| $O_3 + C_3H_6 \rightarrow CH_2OOB + CH_3CHO$ | $2.75\times10^{-15}\times exp(-1880/T)$/molecules $^{-1}$ cm$^3$ s$^{-1}$ | (R5) |
| $O_3 + C_3H_6 \rightarrow CH_3CHOOA + HCHO$ | $2.75\times10^{-15}\times exp(-1880/T)$/molecules $^{-1}$ cm$^3$ s$^{-1}$ | (R6) |
| $O_3 + C_5H_8 \rightarrow CH_2OOE + MACR$ | $3.09\times10^{-15}\times exp(-1995/T)$/molecules $^{-1}$ cm$^3$ s$^{-1}$ | (R7) |
| $O_3 + C_5H_8 \rightarrow CH_2OOE + MVK$ | $2.06\times10^{-15}\times exp(-1995/T)$/molecules $^{-1}$ cm$^3$ s$^{-1}$ | (R8) |
| $O_3 + C_5H_8 \rightarrow HCHO + MACROOA$ | $3.09\times10^{-15}\times exp(-1995/T)$/molecules $^{-1}$ cm$^3$ s$^{-1}$ | (R9) |
| $O_3 + C5H_8 \rightarrow HCHO + MVKOOA$ | $2.06\times10^{-15}\times exp(-1995/T)$/molecules $^{-1}$ cm$^3$ s$^{-1}$ | (R10) |
| $O_3+ HO_2 \rightarrow OH$ | $2.03\times10^{-16}\times (T/300)^{4.57}\times exp(693/T)$/molecules $^{-1}$ cm$^3$ s$^{-1}$ | (R11) |
| $RO_2+ NO_2 \rightarrow$ peroxy nitrates | $(3.28\times10^{-28}\times7.24\times10^{18}\times P/T\times(T/300)^{-6.87}\times1.125$ $\times10^{-11}\times(T/300)^{-1.105})\times10^{(log10(0.30)}/(1+(log10(2.93$ $\times10^{-17}\times7.24\times10^{18}\times P/T\times(T/300)^{-5.765})/0.75-1.27$ $\times log10(0.30))^2))/(2.926\times10^{-17}\times7.24\times10^{18}\times P/T$ $\times(T/300)^{-5.765})$/molecules $^{-1}$ cm$^3$ s$^{-1}$ | (R12) |
| $NO_2+ OH \rightarrow HNO_3$ | $3.2\times10^{-30}\times7.24\times10^{18}\times P/T\times(T/300)^{-4.5}\times3\times10^{-11}$ $\times10^{log10(0.41)}/(1+(log10(3.2\times10^{-30}\times7.24\times10^{18}\times P/T$ $\times(T/300)^{-4.5}/3\times10^{-11})/(0.75-1.27$ $\times(log10(0.41))^2)/(3.2\times10^{-30}\times7.24\times10^{18}\times P/T$ $\times(T/300)^{-4.5}+3\times10^{-11})$/molecules $^{-1}$ cm$^3$ s$^{-1}$ | (R13) |
| $O_3+ OH \rightarrow HO_2$ | $1.70\times10^{-12}\times exp(-940/T)$/molecules $^{-1}$ cm$^3$ s$^{-1}$ | (R14) |

*The rate coefficient obtained from the MCM v3.3.1 model.

512  In total, three-stage simulations were carried out to obtain the 4-min photochemical reactions in

513  the reaction and reference chambers, and all three-stage models were operated in a time-dependent mode

514  with a 1 s resolution. In the $1^{st}$-stage, to establish a real atmospheric environment system, all observations

515  on 7 December 2021, from 6:00-11:30, were used to constrain the model to obtain the mixing ratios of

516  the unmeasured species in the ambient atmosphere, including oxygenated VOCs (OVOCs, in total 16

517  species), and nonmethane hydrocarbons (in total 47 species), $O_3$, NO, $NO_2$, $J$ values, T, RH, and pressure

518  ($P$). Because $O_3$-NO-$NO_2$ was not in a steady state when all species were constrained, we conducted a

519  $2^{nd}$-stage simulation from 11:30–12:00. In this stage, we used the output mixing ratios of the unmeasured

520  species from the simulation in the last 1 s of the $1^{st}$-stage simulation as the input, which were not

521  constrained after providing initial values. For the measured species, $O_3$, NO, and $NO_2$ were no longer

522  constrained after providing initial values, while all other variables (including $NO_X$, VOCs, $J$ values, RH,

523  $T$, $P$, etc.) were still constrained in a time-dependent mode with a 1 s resolution after providing initial

524  values. In the $3^{rd}$-stage, we modeled the 4-min photochemical reactions in the reaction and reference

525  chambers. We used the output mixing ratios of the unmeasured species (i.e., OH, $HO_2$, $RO_2$, $SO_2$, HONO,

526  etc.) from the simulation in the last 1 s of the $2^{nd}$-stage simulation and all measured values (i.e., $O_3$, NO,

527  $NO_2$, VOCs, $J$ values, RH, $T$, $P$, etc.) as the model input, which were not constrained after providing

528  initial values. In addition, while maintaining the setup conditions for the $2^{nd}$-stage of the simulation, we

529  extended the simulation of the environment to 12:04 to obtain the modeled $P(O_3)_{net}$ in the environment

530  in the $3^{rd}$-stage simulation. The result is shown in orange marker in Fig. 10d. Figure 7 shows an explicit

531  explanation of the $3^{rd}$-stage simulation in the reaction and reference chambers.

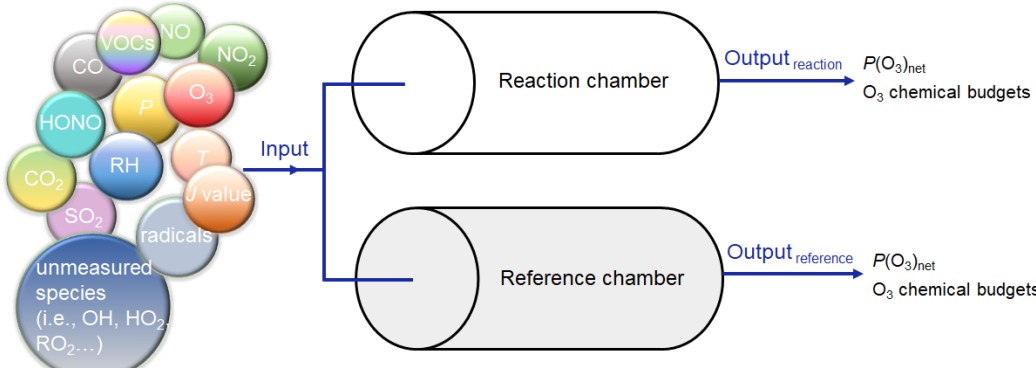


533 **Figure 7: Explicit explanation of the $3^{rd}$-stage model simulation (input meteorological conditions: $P$: 1015.3**
534 **hPa, $T$: 295.6 K, RH: 39.7 %).**

535  Specifically, because the photolysis frequencies play critical roles in the simulation of $P(O_3)_{net}$, the

$J$ values obtained from two methods (labeled methods I and II) were used in the 3rd-stage 4-min
simulation. The $J$ values used in method I were obtained from the measured values (including $J(NO_2)$,
$J(O^1D)$, $J(HONO)$, $J(H_2O_2)$, $J(NO_3\_M)$, $J(NO_3\_R)$, $J(HCHO\_M)$, and $J(HCHO\_R)$) and the simulated values
using the Tropospheric Ultraviolet and Visible (TUV) radiation model (version 5.3) (including $J(HNO_3)$,
$J(CH_3CHO)$, $J(MACR)$, $J(MEK)$, $J(HOCH_2CHO)$, $J(C_2H_5CHO)$, $J(C_3H_7CHO)$, and $J(C_4H_9CHO)$). ), while the $J$
values in method II were all obtained from the simulated values using the TUV model, detailed
information on these two methods is introduced in Appendix IV (Tables S12 and S13, respectively).

**3.2.2 Radical chemistry in the reaction and reference chambers**

The variations in the radical mixing ratios (i.e., $HO_2$, $OH$, $RO_2$) and $NO_3$, $NO$, $NO_2$, and $O_3$ mixing
ratios obtained from method I and method II during the 3rd-stage 4-min model simulation are shown in
Fig. 8 and Fig. S18, respectively. The production and destruction reactions of $HO_2$, $OH$, $RO_2$, and $NO_3$
in the reaction and reference chambers obtained from methods I and II are shown in Fig. 9 and Fig. S19,
respectively, the production and destruction reactions of $RO_X$ in the reaction and reference chambers
obtained from methods I and II are shown in Fig. S20, the detailed ROx production pathways of
$NO_3$+VOCs are shown in Fig. S21, and the final modeling results are shown in Fig. 10 and Fig. S22.
From Fig. 8, in the reaction chamber, the $HO_2$, $OH$, $RO_2$, and $NO_3$ concentrations first slightly
increased and then became stable, and their final concentrations were $2.00 \times 10^8$, $7.64 \times 10^6$, $1.08 \times 10^8$, and
$8.47 \times 10^6$ molecules $cm^{-3}$, respectively. In the reference chamber, the $HO_2$ and $RO_2$ concentrations
dropped during the 1st half minute and rose afterward. The final $HO_2$ concentration ($1.35 \times 10^8$ molecules
$cm^{-3}$) was lower than that in the reaction chamber, while the $RO_2$ concentration exceeded that in the
reaction chamber at the end of the 2nd minute and gradually became stable at $1.27 \times 10^8$ molecules $cm^{-3}$.
The OH concentration dropped significantly at the 1st minute and then became stable at approximately
$6.16 \times 10^5$ molecules $cm^{-3}$. The $NO_3$ concentration rose significantly during the 4-min simulation and
reached $3.55 \times 10^7$ molecules $cm^{-3}$ at the end, which was much higher than that in the reaction chamber.

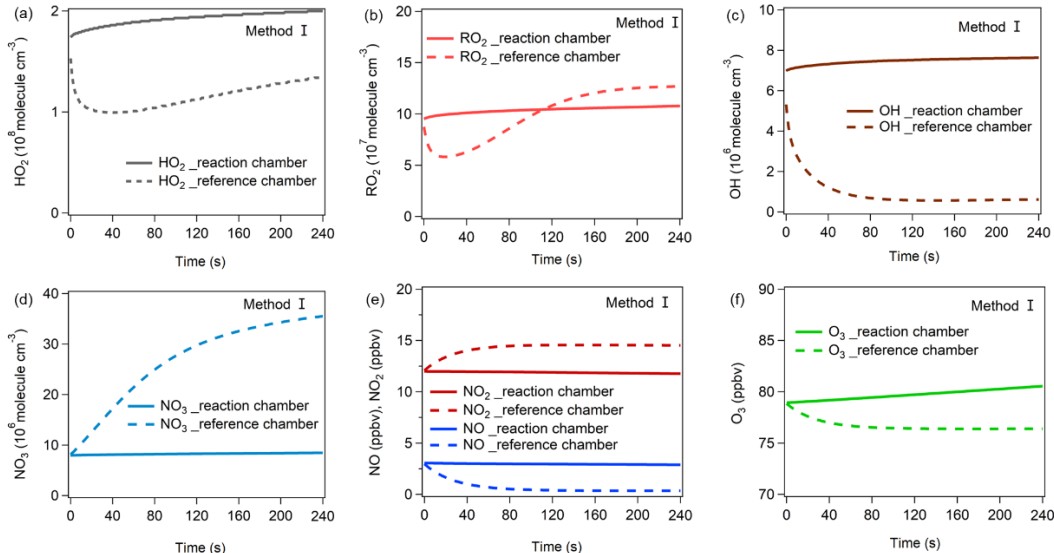


**Figure 8: The variations of (a)$HO_2$, (b) $RO_2$, (c) OH, (d)$NO_3$, (e) NO, $NO_2$, and (f) $O_3$ mixing ratios during the**
**$3^{rd}$-stage 4-min model simulation using method I. The mixing ratios changes of these items for method II is**
**shown in Fig. S18.**

The OH, $HO_2$, $RO_2$, and $NO_3$ concentrations greatly impact the $O_3$ production and destruction rate.

To better understand the factors that drive the OH, $HO_2$, $RO_2$, and $NO_3$ concentration changes, we have
added their production and destruction pathways in Fig. 9. We found that the decrease in $HO_2$ and $RO_2$
concentrations in the reference chamber in the $1^{st}$ half minute was mainly due to NO titration effects, as
high NO mixing ratios existed during the $1^{st}$ half minute. The increase in $HO_2$ concentrations afterward
was largely attributable to $RO+O_2$ reaction/RO decomposition, OH+CO/VOCs reaction, OVOCs
photolysis (i.e., $C_3H_4O_2$, $C_2H_2O_2$, $C_4H_6O_2$), and $NO_3$+VOCs reaction, and the increase in $RO_2$
concentrations afterward were largely attributable to OH+VOCs oxidation, OVOCs photolysis and
$O_3$+VOCs reaction. The main OH sources in the reference chamber were both $HO_2$+NO in method I and
method II. Due to sufficiently high $J(NO_3)$ (~ 90% of that in the reaction chamber) and $NO_2$
concentrations in the reference chamber, the $NO_3$ photolysis and $NO_2$+$NO_3$ reaction consumed $NO_3$ in
the reference chamber, but the $NO_3$ concentrations were still sufficiently high due to high production
rates of $NO_3$ at the same time. The main $NO_3$ source in the reference chamber was the $NO_2$+$O_3$ reaction,
followed by $N_2O_5$ decomposition. The NO concentrations were relatively high in the $1^{st}$ minute and
consumed $NO_3$ very quickly, but due to continuous $NO_3$ sources, the net $NO_3$ production rates ($P(NO_3)_{net}$)
were positive (as shown in Fig. 9), which caused the $NO_3$ concentration to continue to increase (as shown
in Fig. 8d). The main difference in $NO_3$ production in the reference chamber compared to that in the
reaction chamber was the much higher $N_2O_5$ decomposition, which was mainly due to the high $NO_2$
concentrations in the reference chamber. On the other hand, although the $NO+NO_3$ reaction was also one
of the dominant $NO_3$ destruction pathways, $NO_3$ consumed by the $NO+NO_3$ reaction was significantly
smaller than $NO_3$ produced by the $NO_2+O_3$ reaction. Furthermore, in order to check if the $NO_3+VOCs$
reactions exists, we extracted all the $P(ROx)$ pathways related to $NO_3+VOCs$ reactions during the 3rd-
stage 4-min model simulation in the reaction and reference chambers in method I, as shown in Fig. S20.
We found that the $NO_3+VOCs$ reactions are mostly related to the OVOCs (i.e. 6-Ethyl-m-cresol and 3-
Ethyl-6-methylbenzene-1,2-diol) in Fig. S21. The production and destruction rates of ROx are shown in
Fig. S20.

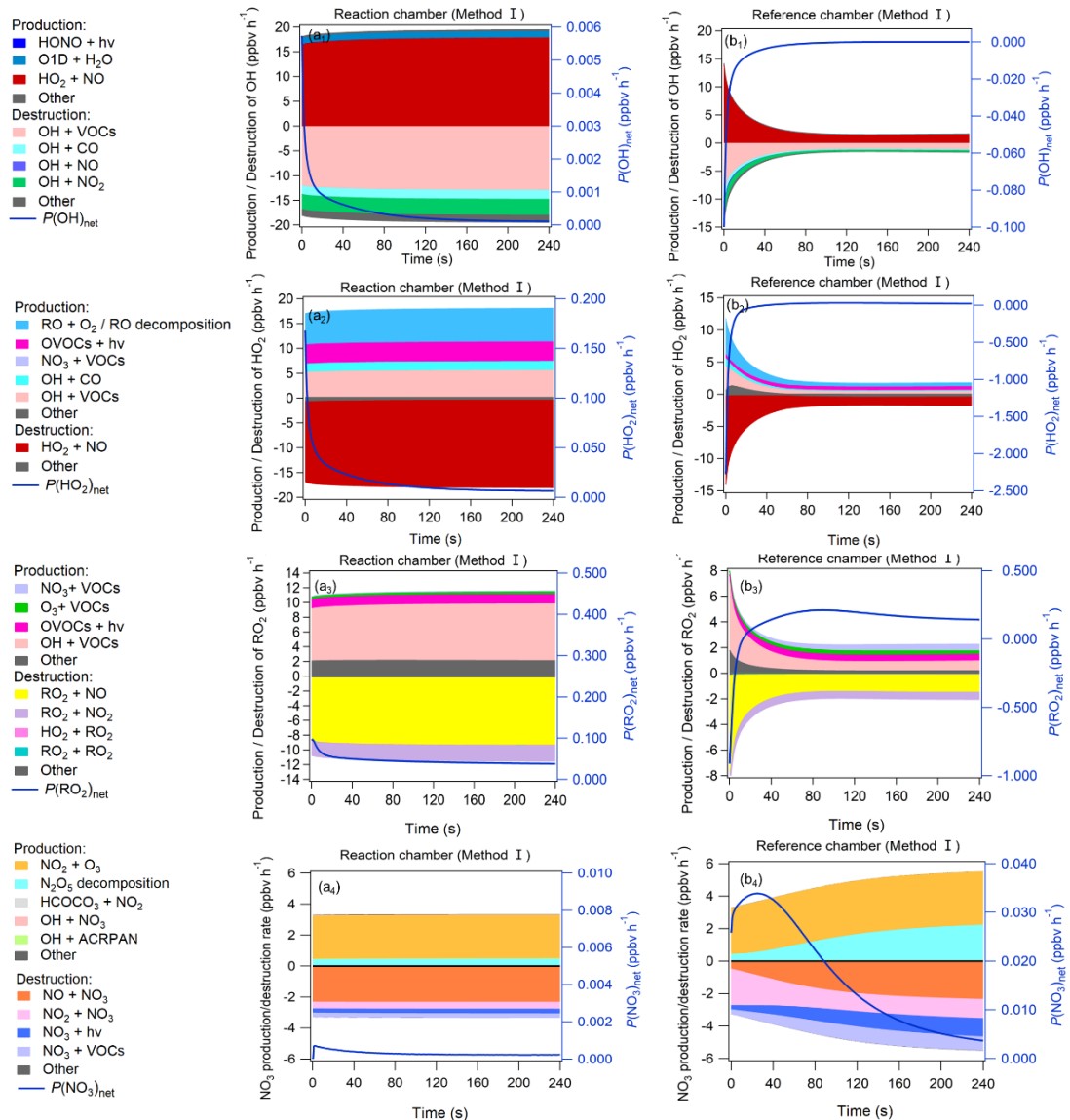

**Figure 9: Production and destruction pathways of OH ($a_1$-$b_1$), HO$_2$ ($a_2$-$b_2$), RO$_2$ ($a_3$-$b_3$), and NO$_3$ ($a_4$-$b_4$) during**
**the 3<sup>rd</sup>-stage 4-min model simulation in the reaction and reference chambers in method I. The related contents**
**for method II (c)-(d) are shown in Fig. S19 in the supplementary materials.**

### 3.2.3 $P(O_3)_{net}$ formation and destruction pathways in the reaction and reference chambers

Figures 10a–d show the modeled $P(O_3)_{net}$ and the sources and sinks of various species during the
3<sup>rd</sup>-stage 4-min simulation. Figure 10a shows the steady state of $P(O_3)_{net}$ and the various species in the
ambient atmosphere achieved in the last 1 s of the 2<sup>nd</sup>-stage simulation; Figures 10b and c show the
modeled $P(O_3)_{net}$ and the $O_3$ chemical budgets in the reaction and reference chambers during the model
simulation period; Figure 10d summarizes the modeled $P(O_3)_{net}$ in the ambient air (represented as blue
and orange markers at the time when the ambient air was going in and out of the NPOPR system,
respectively) and the modeled $P(O_3)_{net}$ in the reaction and reference chambers. To compare the modeled
results with our measured results, we calculated the integral mean of the modeled $P(O_3)_{net}$ in the reaction
and reference chambers and appended the related measured $P(O_3)_{net}$ value during the 4-min simulation
time onto Fig. 10d (green maker). Furthermore, the reaction weights of different production and
destruction reaction processes of $O_3$ are shown in Figs. 10e–h.

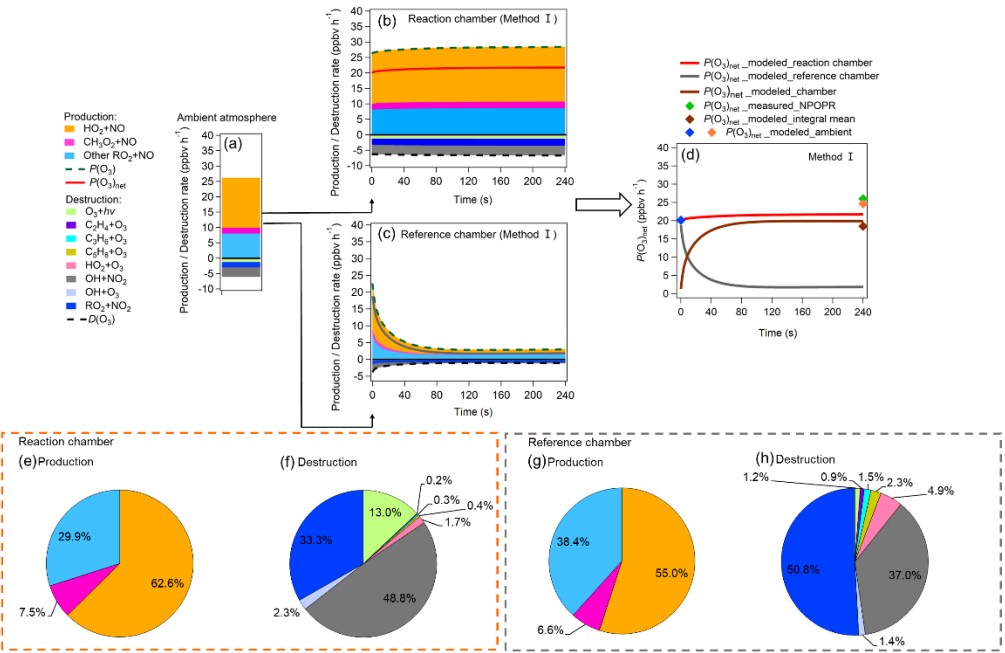


**Figure 10: (a)–(c) Modeled $P(O_3)_{net}$ values and $O_3$ chemical budgets in (a) ambient air when injected into the**
**reaction and reference chambers and (b–c) the reaction and reference chambers during the 4-min model**
**simulation; (d) $P(O_3)_{net}$, where $P(O_3)_{net}$_modeled_ambient represent the modeled $P(O_3)_{net}$ in the ambient air**
**at the time before (blue marker) and after (orange marker) the sampled ambient air was injected into the**
**dual-channel reaction chamber. $P(O_3)_{net}$_modeled_reaction chamber and $P(O_3)_{net}$_modeled_reference**
**chamber represent the $P(O_3)_{net}$ change trends during the 4-min photochemical reactions in the reaction and**
**reference chambers, respectively. $P(O_3)_{net}$_modeled_chamber = $P(O_3)_{net}$_modeled_reaction chamber–**

$P(O_3)_{net}$\_modeled\_reference **chamber, and** $P(O_3)_{net}$\_modeled\_integral **mean represents the integral mean of**

**detection system. (e)–(h) Reaction weights of each production and destruction reaction process of** $O_3$ **in the**

**619**    Figure 10a-h show the contributions of different reaction pathways to $P(O_3)$ and $D(O_3)$. $P(O_3)$ and

**620**    $D(O_3)$ were almost the same within the 4-min reaction in the reaction chamber (all species reached a

**621**    steady-state condition), while $P(O_3)$ and $D(O_3)$ in the reference chamber significantly decreased within

**622**    the 1$^{st}$ min and remained stable in the following minutes. In the reaction chamber, the $HO_2+NO$ reaction

**623**    contributed the most to $P(O_3)$, accounting for 62.6 % of the total $P(O_3)$, with an integral mean value of

**624**    17.5 ppbv h$^{-1}$ in the reaction chamber. The second important pathway of $P(O_3)$ was $RO_2+NO$ (accounting

**625**    for 37.4 % of the total $P(O_3)$). The $RO_2+NO$ core reaction accounts for more than approximately 1200

**626**    types of $RO_2$ radicals, and the pathway of $CH_3O_2+NO$ contributed 7.5 % to the total $P(O_3)$. The most

**627**    important contributor of $D(O_3)$ was $OH+NO_2$ (48.8 %), followed by $RO_2+NO_2$ (33.3 %), $O_3$ photolysis

**628**    (13.0 %), $O_3+OH$ (2.3 %), $O_3+HO_2$ (1.7 %), $C_5H_8+O_3$ (0.4 %), $C_3H_6+O_3$ (0.3 %), and $C_2H_4+O_3$ (0.2 %).

**629**    In the reference chamber, the integral mean value of the $HO_2+NO$ reaction was 2.3 ppbv h$^{-1}$, which had

**630**    the largest contribution to $P(O_3)$ (accounting for 55.0 %). The second largest contributor to $P(O_3)$ was

**631**    $RO_2+NO$ (accounting for 45.0 % of the total $P(O_3)$), in which the pathway of $CH_3O_2+NO$ contributed

**632**    6.6 % to the total $P(O_3)$. The most important contributor of $D(O_3)$ was $RO_2+NO_2$ (50.8 %), followed by

**633**    $OH+NO_2$ (37.0 %), $O_3+HO_2$ (4.9 %), $C_5H_8+O_3$ (2.3 %), $C_3H_6+O_3$ (1.5 %), $O_3+OH$ (1.4 %), $O_3$ photolysis

**634**    (1.2 %), and $C_2H_4+O_3$ (0.9 %). For all $P(O_3)$ reactions, the weight of the $RO_2+NO$ reaction in the

**635**    reference chamber was 7.5 % higher than that in the reaction chamber; however, for all $D(O_3)$ reactions,

**636**    the weight of the $RO_2+NO_2$ reaction in the reference chamber was 17.5 % higher than that in the reaction

**637**    chamber, which mitigated the high $P(O_3)$ value caused by $RO_2+NO$ in the reference chamber.

**638**    Furthermore, the weight of the $OH+NO_2$ reaction in the reference chamber was 11.9 % lower than that

**639**    in the reaction chamber, which may be the main reason that led to $NO_2$ mixing ratios in the reference

**640**    chamber being much higher than those in the reaction chamber. It is worth noting that the different

**641**    reaction pathways of $P(O_3)$ and $D(O_3)$ stabilized at approximately 1.5 min for both methods I and II (as

**642**    shown in Figs. 10 and S22, respectively), and the radicals that play critical roles in photochemical $O_3$

**643**    formation, such as $HO_2$, $RO_2$ and $OH$, reached quasi-steady states in approximately 3 min (as shown in

**644**    Figs. 8 and S18). As the lowest experimental residence time in the reaction and reference chambers was

**645**    3.8 min at an air flow rate of 5 L min$^{-1}$, the photochemical reaction time at different air flow rates in the

NPOPR system was sufficient for $P(O_3)_{net}$ investigation, and it is reasonable for us to set the alternate
ambient air sampling time for the reaction and reference chambers to 2 min, where the ambient air
actually had already reacted for at least 3.8 min in the chambers.

The $P(O_3)_{net}$ value measured by the NPOPR detection system at 12:04 was 26.0 ppbv h$^{-1}$, which is

1.4 ppbv h$^{-1}$ higher than the modeled $P(O_3)_{net}$ value in the ambient air (orange marker in Fig. 10d; 24.6
ppbv h$^{-1}$) and 7.5 ppbv h$^{-1}$ higher than the modeled $P(O_3)_{net}$ value of the NPOPR system (brown marker
in Fig. 10d; 18.5 ppbv h$^{-1}$, as calculated from the integral mean of $P(O_3)_{net}$ in the 3$^{rd}$-stage 4-min
simulation in the reaction and reference chambers). Here, we note that for a better comparison between
the measured and modeled $P(O_3)_{net}$ values, the measured $P(O_3)_{net}$ used here was obtained from a 4-min
time resolution, which is 1.4 ppbv h$^{-1}$ higher than the measured $P(O_3)_{net}$ value used in Fig. 6 (1-h time
resolution). The ratio of the measured and modeled $P(O_3)_{net}$ values was 1.4, which is consistent with the
measured-to-modeled ratio of the cumulative $P(O_3)_{net}$ (1.3 and 1.4) obtained from previous studies
(Cazorla et al., 2012; Ren et al., 2013), where $P(O_3)_{net}$ values were also directly measured in the
atmosphere and were independent of the OH and HO$_2$ measurements. The reason for the difference
between the measured and modeled $P(O_3)_{net}$ here may be due to the inaccurate estimation of HO$_2$/RO$_2$
radicals; for example, Ren et al. (2013) found that $P(O_3)$ calculated from the modeled HO$_2$ was lower
than that calculated from the measured HO$_2$. The unknown HO$_2$ source should be identified for a more
accurate estimation of $P(O_3)_{net}$ in future studies.

Additionally, the modeled $P(O_3)_{net}$ using the $J$ values obtained from method II was 9.3 ppbv h$^{-1}$

lower than the measured $P(O_3)_{net}$, and this discrepancy was slightly larger than that using method I, as
shown in Appendix IV (Fig. S22). The differences in the measured and modeled $P(O_3)_{net}$ by methods I
and II were 28.8 % and 35.8 %, respectively. This difference was mainly due to the transmittance of
$J(NO_2)$ in method II (30 %) being much higher than that in method I (9 %), and NO$_2$ photolysis products
were involved in the main reaction of O$_3$ production through HO$_2$+NO and RO$_2$+NO, so the modeled
$P(O_3)_{net}$ in the reference chamber was slightly overestimated in method II, thus leading to an
underestimation of the final $P(O_3)_{net}$.

Furthermore, because the NO$_2$ data used here were measured by a commercially available

chemiluminescence NOx monitor, the NO$_2$ and NOx mixing ratios could be overestimated due to NOz
interference (i.e., HNO$_3$, peroxyacetyl nitrate (PANs), HONO, etc.) (Dunlea et al., 2007). According to
our results, the chemiluminescence technique could bias $NO_2$ by 5 % compared to the CAPS technique,
which is regarded as a trustworthy $NO_2$ measurement technique without chemical interference. Therefore,
we simulated the interference of $NO_2$ measured by a chemiluminescence NOx monitor in method I as
follows: reducing and increasing the ambient $NO_2$ mixing ratios in the reaction and reference chambers
by 5 % in the $3^{rd}$-stage 4-min simulation. The results showed that increasing and decreasing $NO_2$ by 5 %
resulted in a decrease in $P(O_3)_{net}$ by 1.64 % and 3.68 %, respectively (as shown in Fig. S23), which is
much smaller than the bias caused by $P(O_3)_{net}$ in the reference chamber (which were 13.9 % and 22.3 %
for methods I and II, respectively). To evaluate $P(O_3)_{net}$ error caused by the measurement error of $J$ values,
we introduced a ±5 % error to the measured $J$ values during the $3^{rd}$ stage of the 4-min simulation in
method I. The modeled $P(O_3)_{net}$ results are presented in Fig. S24 in the supplementary materials. We
observed that the inclusion of a -5 % measurement error in $J$ values led to a decrease in $P(O_3)_{net}$ by
7.27 %, while adding a +5 % measurement error in $J$ values caused an increase in $P(O_3)_{net}$ by 3.08 %.
This implies that the maximum bias of $P(O_3)_{net}$ caused by the measurement error of $J$ values falls within
the error range of the currently assessed $P(O_3)_{net}$ error, which was 13.9 % for method I. Therefore, we
conclude that this type of error will not influence our final modeling results and conclusions.
In conclusion, modeling tests demonstrated that the radicals and gas species in the reaction
chamber of the NPOPR detection system were similar to those in genuine ambient air, while these
radicals also unexpectedly existed in the reference chamber. This was mainly because the UV protection
film used by the reference chamber did not completely filter out sunlight, which led to the low
transmittance of light ranging from 390 nm to 790 nm. The $P(O_3)_{net}$ biases caused by this interference
modeled in methods I and II were 13.9 % and 22.3 %, respectively, which ensured that the measured
$P(O_3)_{net}$ by the NPOPR detection system should be regarded as the lower limit values of real $P(O_3)_{net}$ in
the atmosphere. We recommend that the $J$ values obtained from method I should be used in the model
simulation, which can better explain the photochemical formation of $O_3$ in the actual atmosphere, but if
direct $J$ value measurements cannot be achieved during field observations, the $J$ values obtained from
method II would also be acceptable in modeling studies.
**4 Conclusions**
We modified and improved a net photochemical ozone production rate (NPOPR) detection system based
on a dual-channel reaction chamber technique, which provides more accurate results and has broader
application potential compared to previous studies. The main improvements of NPOPR detection system
compared to previous studies were as follows: (1) improved the design of the reaction and reference
chambers to make sure they have good airtightness; (2) changed the air sampling structure to enable the
total air flow rates change freely from 1.3 to 5 L min$^{-1}$ in the reaction and reference chambers, which can
make the NPOPR system achieve different limits of detection (LODs) and appliable to different ambient
environment; (3) characterized the NPOPR detection system at different air flow rates to optimize the
$P(O_3)_{net}$ measurements, the LODs of the NPOPR detection system are 0.07, 1.4, and 2.3 ppbv h$^{-1}$ at air
flow rates of 1.3, 3, and 5 L min$^{-1}$, respectively; (4) tested the performance of both reaction and reference
chambers by combining the field measurement and the MCM modeling method.

The NPOPR detection system was employed in the field observation at the Shenzhen

Meteorological Gradient Tower (SZMGT), which is located in PRD, China. During the measurement
period, the $P(O_3)_{net}$ was around zero during nighttime and ranged from ~ 0 to 34.1±7.8 ppbv h$^{-1}$ during
daytime (from 6:00–18:00), with the average value of 12.8 (±5.5) ppbv h$^{-1}$. Besides, $P(O_3)_{net}$ start to
increase at around 7:00 at local time, this may be due to the rise of the $O_3$ precursors (i.e., VOCs)
transported down from the high-altitude atmospheric residual layer to the near-surface and the increase
of solar radiation intensity increased the atmospheric oxidation capacity. $P(O_3)_{net}$ was then reaches a peak
at around 12:00 at noon time, by coupling with diurnal $O_3$ mixing ratios trends, we confirmed that the
ground-level $O_3$ mixing ratios were influenced by both photochemical production and physical transport.

In order to clarify the detailed photochemical reaction processes in the reaction and reference

chambers of NPOPR system, we modeled the $P(O_3)_{net}$ on 7 December 2021, 12:00-12:04 in the reaction
and reference chambers using MCM v3.3.1. As the photolysis frequencies of different species ($J$ values)
play critical roles in the formation of $P(O_3)_{net}$, the $J$ values obtained from two methods were used in the
4-min chamber photochemical reaction (labeled as method I and method II), in method I , eight main $J$
values (e.g., $J(NO_2)$, $J(O^1D)$, $J(HONO)$, etc.) were measured directly, and other $J$ values were obtained
from the simulated values using the Tropospheric Ultraviolet and Visible (TUV) radiation model, while
in method II, $J$ values were all obtained from the simulated values using TUV model (as described in
Sect. 3.2). Modeling tests demonstrated that the mixing ratios of different radicals and gas species (i.e.,
OH, HO$_2$, RO$_2$, NO$_3$, NO, NO$_2$, and O$_3$) in the reaction chamber were similar with those in the real

ambient environment, while due to the UV protection film used by the reference chamber does not completely filter out the sunlight, there was low transmittance of the light ranged from 390 nm to 790 nm. In the reaction chamber, the contribution of different reactions to $P(O_3)$ and $D(O_3)$ modeled by method I and II were quite similar, where the $HO_2+NO$ reaction contributed most to $P(O_3)$ (~ 62.6 %), followed by the $RO_2+NO$ reaction (~ 37.4 %). The $OH+NO_2$ reaction contributed most to $D(O_3)$, which accounted for ~ 48.9 %, followed by the $RO_2+NO_2$ reaction $O_3$ photolysis, which accounted for ~ 33.3 % and 13.0 %, respectively. In the reference chamber, the contribution of different reactions to $P(O_3)$ and $D(O_3)$ modeled by method I and II were different, where the $HO_2+NO$ reaction contributed ~ 55.0 % and ~ 58.2 % to the total $P(O_3)$, respectively, and $RO_2+NO$ contributed ~ 44.9 % and 41.8 % to the total $P(O_3)$, respectively. The most important contributor of $D(O_3)$ modeled by method I was $RO_2+NO_2$ (50.8 %), followed by $OH+NO_2$ (37.0 %), while the most important contributor of $D(O_3)$ modeled by method II was $OH+NO_2$ (46.8 %), followed by $RO_2+NO_2$ (44.1 %). For all $P(O_3)$ reactions, the weight of $RO_2+NO$ reaction in the reference chamber was 7.5 % and 4.3 % higher than that in the reaction chamber in method I and II, respectively, however, for all $D(O_3)$ reactions, the weight of $RO_2+NO_2$ reaction in the reference chamber was 17.5 % and 10.9 % higher than that in the reaction chamber in method I and II, respectively, which will somehow mitigate the high $P(O_3)$ caused by $RO_2+NO$ in the reference chamber. The different reaction pathways of $P(O_3)$ and $D(O_3)$ had stabilized at around 1.5 min, and the radicals that play critical roles in photochemical $O_3$ formation, such as $HO_2$, $RO_2$ and $OH$, reached quasi-steady states in about 3 min, the long enough ambient air residence time in the reaction and reference chambers ($\geq$ 3.8 min) make the photochemical reaction time at different air flow rates in the NPOPR system sufficient enough for investigating the $P(O_3)_{net}$, and it is reasonable for us to set the alternate ambient air sampling time for the reaction and reference chambers at 2 min, where the ambient air actually has already reacted for at least 3.8 min in the chambers.

The biases of the modeled $P(O_3)_{net}$ caused by the interference of the reactions in the reference chamber in methods I and II were 13.9 % and 22.3 %, respectively; thus, the measured $P(O_3)_{net}$ by the NPOPR detection system should be regarded as the lower limit values of the real $P(O_3)_{net}$ in the atmosphere. Nevertheless, the measured $P(O_3)_{net}$ values were 7.5 and 9.3 ppbv h$^{-1}$ higher than the modeled $P(O_3)_{net}$ values obtained from methods I and II, respectively, which may be due to the inaccurate modeling of $HO_2/RO_2$ radicals. Short-lived intermediate measurements coupled with direct $P(O_3)_{net}$

measurements are needed in future study in order to studies to better understand the photochemical production and destruction mechanisms of $O_3$. We recommend that the $J$ values obtained from method I should be used in the model simulation, which can better explain the photochemical formation of $O_3$ in the actual atmosphere, but if direct $J$ value measurements cannot be achieved during field observations, the $J$ values obtained from method II would also be acceptable in modeling studies.

The self-built NPOPR detection system in this study filled the gap in the observation method in China. The research results not only help us to better understand the tropospheric $O_3$ budget but also provide an important data basis for formulating correct $O_3$ pollution prevention measures and control strategies.

*Data availability.* The observational data used in this study are available from corresponding authors upon request (junzhou@jnu.edu.cn).

*Author contributions.* JZ, BY, and MS designed the experiment, JZ and YXH developed and assembled the NPOPR detection system, YXH, JZ, JPZ, BY, YW, YFW, SCY, YWP, JPQ, XJH, XS and YBC collected and analyzed the data, YXH and JZ wrote the manuscript, all authors revised the manuscript.

*Competing interests.* The authors declare that they have no known competing interests.

*Acknowledgements.* This study was funded by the Natural Science Foundation of Guangdong Province (grant no. 2020A1515110526), and the Key-Area Research and Development Program of Guangdong Province (grant no. 2020B1111360003). We thank Nan Ma and Xiaofeng Su in Jinan University for the HONO production experiment.

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
