# Peer review of "Measuring and modeling investigation of the Net"

_Atmospheric Chemistry and Physics, 2022_

## Referee Comment (RC2)

**Review of Hao et al., 2022:** "Measuring and modelling investigation of the Net Photochemical Ozone Production Rate via an improved dual-channel reaction chamber technique"

**General Comments:**

This manuscript details changes made to a two-chamber system used to directly measure ambient net ozone production rates, sensitivity tests conducted on this system, and a final 0-D box modeling comparison of the chemistry in both chambers and in ambient air for a select measurement day in 2021. The criticality of making such direct P(O3)net measurements in areas with poor air quality for the purpose of determining efficient $O_3$ mitigation strategies is not refuted here; this is an important issue globally. However, while it is clear that much work has gone into the manuscript, it requires major revisions both scientifically and grammatically before being referred for publication. Given the nature of the manuscript, I wonder whether submission to Atmospheric Measurement Techniques is more suitable.

**Specific Comments:**
- In general, the manuscript is structured appropriately, but has a number of grammatical errors within that prevent clear and fluent presentation. The manuscript would benefit from a review by a technical editor. Some suggestions are presented in the technical corrections below.
- Equation (1),  the ambient $O_3$ budget,  is also dependent upon entrainment from the stratosphere. Please add this term for completeness.
- The manuscript indicates that the same tests were conducted on the NPOPR system as in other studies (L188-L189). However, several sensitivity tests conducted in previous studies (Sklaveniti et al., 2018; Baier et al., 2017) are missing from this analysis. These include the following: biases on P(O3)net from temperature differences between the reaction and reference chamber; short-duration baseline drifting in the CAPS monitor (in presence of both dried and humidified air) and HONO production. Given that some of these additional tests have not been conducted, it is difficult to assess whether the NPOPR system described here has significant improvements over other systems described in the literature. Further, a more detailed description of the system, materials used,  switching for sample analysis, etc. would be helpful for characterizing potential differences between this chamber system and others described in previous literature.
- A full error quantification and discussion of P(O3)net is missing from the manuscript's main text. Results are presented for several sensitivity tests in the main text, while results from others presented in the supplemental information. However, not all test results that influence the overall P(O3)net_error are included in the error budget for P(O3)net. This includes the loss of $O_x$ in each chamber, the photo-enhanced $O_x$ loss, residence time uncertainty, etc.. A limit of detection is presented in Section 2.4, but is this then later revised in Section 3.2? Each error term in the P(O3)net_error equation should be clearly defined.
- The potential for $O_x$ loss at high RH and photo-enhanced $O_x$ loss was discussed in supplemental information and a correction factor was devised to exclude this bias, but it is unclear how this correction factor is (or can be) used for ambient measurements; b) which RH range (30-32% is shown, but ambient RH routinely exceeds this value) was tested; c) for which flow rate regime was this correction devised and d) whether this relationship holds for all measurement scenarios. More details are needed here to fully understand the test being conducted and how it can be applied to correct P(O3)net.
- It is unclear what new information is being presented with the discussion of flow state in L282-L290 and I might suggest that this section could be removed. While the flow may be

considered laminar, it is shown from residence time testing that the fluid within the chambers does not represent a "plug" flow, so reactions on the chamber walls may still produce biases in P(O3)net.

- Units of ppbv imply a "mixing ratio" in atmospheric chemistry terms, whereas a "concentration" is often referred to in molec cm$^{-3}$ or mol L$^{-1}$
- Have the authors investigated the potential for a small amount of $NO_2$ impurity in the NO mixture used for conversion and can you address this potential bias in P(O3)net?
- L330-339 addresses the fact that $O_x$(= $NO_2$+$O_3$) mixing ratios change over the 2 minute sampling time of each chamber. How do the authors address issues of $NO_2$ atmospheric variability over the sampling time of the reaction versus the reference chamber, and the subtraction of these alternating two measurements, especially in urban areas? It is difficult to understand the resolution of P(O3)net in Figure 6 and how the data are averaged (or not?). Can you provide more details on how the data are processed to help the discussion?
- The modeling section could use some work to clarify and condense the information presented. In theory, modeling of the chemistry in both chambers separately seems like a good check on what is actually measured by the NPOPR system, but it is known from previous studies (and this issue is presented here as well) that modeling of $HO_2$ and $RO_2$ in ambient air does not match that which is measured (e.g. from Ren et al. 2013). Thus, if $HO_2$ and $RO_2$ are not well-captured in the model from parameterized VOCs and reactions therein, it is difficult to use the model to verify the chamber chemistry. Therefore, one suggestion could be to simplify this discussion to compare the ambient photochemistry from the model to the NPOPR system, given modeling limitations.

**Technical Corrections:**

- L87: You may also choose to reference Baier et al. (2017) here for similar work
- L90: There is no Baier et al. (2021) in references. Do you mean Baier et al. (2015)?
- L98: Could also benefit from citing Baier et al., 2017 for chamber artifact discussion
- L107-108: please re-write for clarity
- L116: change "access" to assess
- L131: change "amounted" to mounted
- L132: Please describe alternating flow more clearly. How do you account for the "transition" period after switching and sampling by the CAPS monitor (i.e. is some portion of the data discarded after transitioning between reaction and reference chambers as in Sklaveniti et al., 2018?)
- L141: How do you assess portability?
- L186: Accuracy implies that P(O3) is produced as it is in the atmosphere, which cannot be determined here. Better phrasing could be: "to make the NPOPR less prone to biases than other systems" or similar.
- L217: Please calculate the bias in P(O3)net incurred from $O_x$ loss in the reaction and reference chamber
- Table 1: is some shading missing here?
- Figure 4: Typically, the axes are reversed for a calibration such that y represents known values and x represents those values that are calibrated.
- L360: Please describe what 'cd' means for readers
- Figure 5: Please indicate shading in a colorbar, etc.
- Figure 6b): is this an average of the diurnal P(O3)
- L432: November or December?
- SI L129: perhaps change the word "ingestion" to "loss?
- SI L135-136 should be Fig. S7a, not S6a
- Figure S7: please add units on figure axes

- Table S7a: Sklaveniti et al., 2018 does not constitute an urban area – please see description within this particular reference and check other sites as well.
- SI Tables S7a/S7b require corrections made to locations. For example, Writtle College is not located in the USA; Houston, USA should be replaced with City, State, Country format like other sites: Houston, Texas, USA, etc.

---

## Author Comment (AC1)

We thank the referee for the insightful comments and suggestions, which give us new perspectives to describe our work and greatly helped to improve the significance of the paper. Our answers are listed in the following in red, after the referee's comments, which are in black. The modifications in the text are marked in yellow.

**Reviewer #1**

This paper describes an improved direct measurement system of the photochemical ozone production rate ($P(O_3)$). The authors also performed a field observation of $P(O_3)$ and evaluated the instrument and observation results using a detailed box model simulation.

$O_3$ pollution is a crucial problem for atmospheric environment, and a behavior of $O_3$ is very difficult, so direct measurements of $P(O_3)$ such as this research are very important and valuable. However, this paper has major concerns about the evaluation by the box model simulation. So, I cannot recommend this paper to be published in Atmospheric Chemistry and Physics in the present form. The authors should resolve the concerns by re-evaluation of the box model simulation or other ways.

Thank you very much for providing useful suggestions. We agree with the reviewer that some of the modeling results in the reference chamber were beyond our expectation. Therefore, we have re-evaluated and carefully checked each step of the box model simulations described in Sect. 3.2, but no error has been found so far. To make the modeling results clearer, we added the budgets of OH, $HO_2$, $RO_2$, and $NO_3$ during the $3^{rd}$-stage 4-min model simulation in the reaction and reference chambers. The main purposes of performing the box model simulation in this study are as follows: ① how to determine the photolysis frequencies for $P(O_3)_{net}$ modeling in ambient air by comparing the modeling results with the measured results, and ② provide a comprehensive understanding of the radical budgets in the reaction and reference chambers. The reactions occurring in the reference chamber are much more complex than those in the reaction chamber, and the results are also different when we use different $J$ value determination methods (labeled as method I and method II). However, the biases of the modeled $P(O_3)_{net}$ caused by the interferences in the reference chamber were 13.9 % and 22.3 % in method I and method II, respectively, which ensures that the measured $P(O_3)_{net}$ by the NPOPR detection system should be regarded as the lower limit values of the real $P(O_3)_{net}$ in the atmosphere. We made some additional box model simulations and answered the reviewer's questions point by point as follows:

Major comments:

- The results of the box model simulation

I suspect such high concentrations of $NO_3$ in the reference chamber. First, the value of $JNO_3$ in the reference chamber is about 90% of that in the reaction chamber, which is sufficiently high. Second, the rate constant of the reaction of $NO_3$ with NO is sufficiently large and the lifetime of $NO_3$ (at 298 K) is 1.6 s and very short in the presence of NO of 1 ppbv. There might be high concentrations of $NO_3$ if there are very large sources of $NO_3$, but in that case, the authors should mention the evidence. I think the $NO_2 + O_3$ reaction cannot be a large source of $NO_3$. I'm not sure about $N_2O_5$, but it is unlikely that there could be high concentrations of $N_2O_5$ under the temperature during the observation period.

To explore the reason for such a high concentration of $NO_3$ in the reference chamber, we modeled the production and destruction pathways of $NO_3$ in the reaction and reference chambers (as shown in Fig. S19 for method I and method II). First, we found that even when the $J(NO_3)$ values were sufficiently large (which were 90% and 100% of that in the reaction chamber for method I and method II, respectively), which led to a high $NO_3$ photolysis reaction in the reference chamber, the $NO_3$ concentrations were still sufficiently high due to the high production rates of $NO_3$ at the same time (the main production pathway of $NO_3$ is the $NO_2+O_3$ reaction, followed by $N_2O_5$ decomposition). Second, the NO concentrations were large in the 1st minute, which consumes $NO_3$ very quickly, but as there were continuous $NO_3$ sources, the net $NO_3$ production rates ($P(NO_3)_{net}$) were positive, which caused the $NO_3$ concentration to continue to increase (as shown in Figs. 8d and S20).

The main difference of $NO_3$ production in the reference chamber compared to that in the reaction chamber is the much higher $N_2O_5$ decomposition. A proposed mechanism for the decomposition of $N_2O_5$ is as follows (Kotz et al., 2019):

$$N_2O_5 \xrightarrow{k1} NO_2+NO_3 \qquad \text{step 1 (slow)}$$

$$NO_2+NO_3 \xrightarrow{k2} NO_2 + O_2 + NO \qquad \text{step 2 (fast)}$$

$$NO+N_2O_5 \xrightarrow{k3} 3NO_2 \qquad \text{step 3 (fast)}$$

We proposed that due to the much higher $NO_2$ concentrations in the reference chamber than in the reaction chamber, $NO_3$ is consumed to produce NO in step 2, which accelerates step 1 and benefits step 3, thus increasing the steady-state concentration of $NO_3$. This is also demonstrated by the increased $N_2O_5$ decomposition and $NO_2$ concentrations along with the increased $NO_2+NO_3$ reaction (as shown in Figs. 8-9 and S18-S19). Therefore, the high $NO_3$ concentrations in the reference chamber were mainly due to the high $NO_2$ concentrations. On the other hand, although the $NO_2+NO_3$ reaction was also one of the dominant $NO_3$ destruction pathways, $NO_3$ consumed by the reaction of

NO$_2$+NO$_3$ was significantly smaller than NO3 produced by the reaction of NO2+O3 (as shown in Figs. 9 and S19).

We added the related discussion for method I in pages 24-25, lines 551-567 in the modified manuscript:

"Due to sufficiently high $J$(NO$_3$) (~ 90% of that in the reaction chamber) and NO$_2$ concentrations in the reference chamber, the NO$_3$ photolysis and NO$_2$+NO$_3$ reaction consumed NO$_3$ in the reference chamber, but the NO$_3$ concentrations were still sufficiently high due to high production rates of NO$_3$ at the same time. The main NO$_3$ source in the reference chamber was the NO$_2$+O$_3$ reaction, followed by N$_2$O$_5$ decomposition. The NO concentrations were relatively high in the 1$^{st}$ minute and consumed NO$_3$ very quickly, but due to continuous NO$_3$ sources, the net NO$_3$ production rates ($P$(NO$_3$)$_{net}$) were positive (as shown in Fig. 9), which caused the NO$_3$ concentration to continue to increase (as shown in Fig. 8d). The main difference in NO$_3$ production in the reference chamber compared to that in the reaction chamber was the much higher N$_2$O$_5$ decomposition, which was mainly due to the high NO$_2$ concentrations in the reference chamber. On the other hand, although the NO+NO$_3$ reaction was also one of the dominant NO$_3$ destruction pathways, NO$_3$ consumed by the NO+NO$_3$ reaction was significantly smaller than NO$_3$ produced by the NO$_2$+O$_3$ reaction. Furthermore, in order to check if the NO$_3$+VOCs reactions exists, we extracted all the $P$(ROx) pathways related to NO$_3$+VOCs reactions during the 3$^{rd}$-stage 4-min model simulation in the reaction and reference chambers in method I, as shown in Fig. S20. We found that the NO$_3$+VOCs reactions are mostly related to the OVOCs (i.e. 6-Ethyl-m-cresol and 3-Ethyl-6-methylbenzene-1,2-diol) in Fig. S21. The production and destruction rates of ROx are shown in Fig. S20."

And added the related discussion for method II in page 25, lines 330-343 in the modified supplementary materials:

Due to sufficiently high $J$(NO$_3$) (~ 100 % of that in the reaction chamber) and NO$_2$ concentrations in the reference chamber, the NO$_3$ photolysis and NO$_2$+NO$_3$ reaction consumed NO$_3$ in the reference chamber, but the NO$_3$ concentrations were still high due to high production rates of NO$_3$ at the same time. Similar with the results obtained from method I as described in the main manuscript, for method II, the main NO$_3$ source in the reference chamber was the NO$_2$+O$_3$ reaction, followed by N$_2$O$_5$ decomposition. The NO concentrations were relatively high in the 1$^{st}$ minute and consumed NO$_3$ very quickly, but due to continuous NO$_3$ sources, the net NO$_3$ production rates ($P$(NO3)$_{net}$) were positive (as shown in Fig.

S19b4), which caused the NO$_3$ concentration to continue to increase (as shown in Fig. S18d). The main difference in NO$_3$ production in the reference chamber compared to that in the reaction chamber was the much higher N$_2$O$_5$ decomposition, which was mainly due to the high NO$_2$ concentrations in the reference chamber. On the other hand, although the NO+NO$_3$ reaction was also one of the dominant NO$_3$ destruction pathways, NO$_3$ consumed by the NO+NO$_3$ reaction was significantly smaller than NO$_3$ produced by the NO$_2$+O$_3$ reaction. The integrated production and destruction rates of ROx are shown in Fig. S20.

[Figure]

**Figure 9: Production and destruction pathways of OH ($a_1$-$b_1$), HO$_2$ ($a_2$-$b_2$), RO$_2$ ($a_3$-$b_3$), and NO$_3$ ($a_4$-$b_4$) during the 3$^{rd}$-stage 4-min model simulation in the reaction and reference chambers in method I. The related contents for method II (c)-(d) are shown in Fig. S19 in the supplementary materials.**

And added Fig. S19 in the modified supplementary materials:

"Due to sufficiently high $J$(NO$_3$) (~ 100 % of that in the reaction chamber) and NO$_2$ concentrations in the reference chamber, the NO$_3$ photolysis and NO$_2$+NO$_3$ reaction consumed NO$_3$ in the reference chamber, but the NO$_3$ concentrations were still high due to high production rates of NO$_3$ at the same time. Similar with the results obtained from method I as described in the main manuscript, for method II, the main NO$_3$ source in the reference chamber was the NO$_2$+O$_3$ reaction, followed by N$_2$O$_5$ decomposition. The NO concentrations were relatively high in the 1$^{st}$ minute and consumed NO$_3$ very quickly, but due to continuous NO$_3$ sources, the net NO$_3$ production rates ($P$(NO$_3$)$_{net}$) were positive (as shown in Fig. S19b4), which caused the NO$_3$ concentration to continue to increase (as shown in Fig. S18d). The main difference in NO$_3$ production in the reference chamber compared to that in the reaction chamber was the much higher N$_2$O$_5$ decomposition, which was mainly due to the high NO$_2$ concentrations in the reference chamber. On the other hand, although the NO+NO$_3$ reaction was also one of the dominant NO$_3$ destruction pathways, NO$_3$ consumed by the NO+NO$_3$ reaction was significantly smaller than NO$_3$ produced by the NO$_2$+O$_3$ reaction."

[Figure]

**Figure S19: Production and destruction pathways of OH(a₁-b₁), HO₂(a₂-b₂), RO₂(a₃-b₃), and NO₃(a₄-b₄) during the 3ʳᵈ-stag 4-min model simulation in the reaction and reference chambers in method II (c)-(d).**

On the other hand, I also suspect the reaction of $NO_3$ with VOCs as a major source of $RO_2$ in the reference chamber. The rate constants of the reactions of $NO_3$ with VOCs are not so large, and it is questionable that $NO_3$ and $RO_2$ concentrations change in minutes by the reactions of $NO_3$ with VOCs unless there are extremely high concentrations of VOCs.

The reviewer is correct. We actually described it incorrectly here, and we apologize for this mistake. The major $RO_2$ sources in the reference chamber were OH+VOC oxidation, followed by OVOC photolysis (i.e., $C_3H_4O_2$, $C_2H_2O_2$, $C_4H_6O_2$, etc.) and $O_3$+VOC reactions. To better understand the radical chemistry, we included the

production and destruction pathways of OH, $HO_2$, $RO_2$, and $NO_3$ during the 3rd-stage 4-min model simulation in the reaction and reference chambers in Figs. 9 and S19 (as shown above) and moved the production and destruction pathways of ROx to Fig. S20.

[Figure]

**Figure S20: Production and destruction pathways of ROx during the 3rd-stage 4-min model simulation in the reaction and reference chambers. (PAN: Peroxyacetyl Nitrate; PNs: formations of all peroxynitrate (including $CH_3O_2NO_2$ and PAN; X: PAN and the net loss of OH+NO to form HONO (usually small)).**

Accordingly, we modified the related description in page 24, lines 542-552 in the modified manuscript:

"OH, HO$_2$, RO$_2$, and NO$_3$ concentrations greatly impact the O$_3$ production and destruction rate. To better understand the factors that drive the OH, HO$_2$, RO$_2$, and NO$_3$ concentration changes, we have added their production and destruction pathways in Fig. 9. We found that the decrease in HO$_2$ and RO$_2$ concentrations in the reference chamber in the 1$^{st}$ half minute was mainly due to NO titration effects, as high NO mixing ratios existed during the 1$^{st}$ half minute. The increase in HO$_2$ concentrations afterward was largely attributable to RO+O$_2$ reaction/RO decomposition, OH+CO/VOCs reaction, OVOCs photolysis (i.e., C$_3$H$_4$O$_2$, C$_2$H$_2$O$_2$, C$_4$H$_6$O$_2$), and NO$_3$+VOCs reaction, and the increase in RO$_2$ concentrations afterward were largely attributable to OH+VOCs oxidation, OVOCs photolysis and O$_3$+VOCs reaction. The main OH sources in the reference chamber were both HO$_2$+NO in method I and method II."

And in pages 24-25, lines 318-329 in the modified supplementary materials:

"OH, HO$_2$, RO$_2$, and NO$_3$ concentrations greatly impact the O$_3$ production and destruction rate. To better understand the factors that drive the OH, HO$_2$, RO$_2$, and NO$_3$ concentration changes in method II, we have added their production and destruction pathways in Fig. S19. We found that the decrease in HO$_2$ and RO$_2$ concentrations in the reference chamber in the 1$^{st}$ half minute was mainly due to NO titration effects, as high NO mixing ratios existed during the 1$^{st}$ half minute. The HO$_2$ and RO$_2$ concentrations were became stable afterwards, the main production pathway for HO$_2$ was RO+O$_2$ reaction/RO decomposition, followed by OH+ VOCs reaction, OVOCs photolysis (i.e., C$_3$H$_4$O$_2$, C$_2$H$_2$O$_2$, C$_4$H$_6$O$_2$), and NO$_3$+VOCs reaction; while the main production pathway for RO$_2$ was OH+ VOCs reaction, followed by OVOCs photolysis (i.e., C$_3$H$_4$O$_2$, C$_2$H$_2$O$_2$, C$_4$H$_6$O$_2$), OH+CO, NO$_3$+VOCs reaction, etc.; the main destruction pathways for HO$_2$ and RO$_2$ were HO$_2$+NO and RO$_2$+NO, respectively. The main OH production and destruction pathways in the reference chamber was HO$_2$+NO reaction and OH+ VOCs reaction, respectively."

Furthermore, in order to check if the NO$_3$+VOCs reactions exists, we extracted all the $P$(ROx) pathways related to NO$_3$+VOCs reactions during the 3$^{rd}$-stage 4-min model simulation in the reaction and reference chambers in method I, as shown in Fig. S20. We found that the NO$_3$+VOCs reactions are mostly related to the OVOCs (i.e. 6-Ethyl-m-cresol and 3-Ethyl-6-methylbenzene-1,2-diol), as shown in Fig. S21.

The related explanations are added in page 25, line 562-567 in the modified manuscript:

"Furthermore, in order to check if the NO₃+VOCs reactions exists, we extracted all the $P$(ROx) pathways related to NO₃+VOCs reactions during the 3ʳᵈ-stage 4-min model simulation in the reaction and reference chambers in method I, as shown in Fig. S20. We found that the NO₃+VOCs reactions are mostly related to the OVOCs (i.e. 6-Ethyl-m-cresol and 3-Ethyl-6-methylbenzene-1,2-diol) in Fig. S21. The production and destruction rates of ROx are shown in Fig. S20."

And in page 27, lines 354-357 in the modified supplementary materials:

[Figure]

Figure S21: The $P$(ROx) pathways related to NO₃+VOCs reactions during the 3ʳᵈ-stage 4-minute model simulation in the reaction and reference chambers in method I (a)-(b) and method II (c)-(d).

For Figs. 9(b) and S10(b), photodissociation of OVOCs is a significant part of ROx sources in the reference chamber. Why such a large fraction in the absence of UV? The authors should mention the evidence.

The reference chamber was covered by a UV protection Ultem film (SH2CLAR, 3M, Japan) to block the sunlight with the wavelengths < 390 nm, so the film has the transmission of lights > 390 nm, thus the photolysis frequencies of some species still existed, as shown in Tables S7 and S13. Therefore, we still observed OVOCs photodissociation in the reference chamber., i.e., $C_3H_4O_2$, $C_2H_2O_2$, $C_4H_6O_2$ and $C_4H_4O_4$, etc. According to MCMv3.3.1, there are in total 979 OVOCs species photolysis in the reference chamber, some of which are listed as follows:

Photolysis reactions from IUPAC Task Group on Atmospheric Chemical Kinetic Data Evaluation (http://iupac. pole-ether.fr/):

1. $C_3H_4O_2$ (Methylglyoxal) $\xrightarrow{hv:\ 225\text{-}410\ nm}$ $CH_3CO_3$ +CO+$H_2O$ IUPAC (2006)

2. ① $C_2H_2O_2$ (Glyoxal) $\xrightarrow{hv:\ 230.5\text{-}462.0\ nm}$ CO+CO+$H_2$  IUPAC(2006)

   ② $C_2H_2O_2$(Glyoxal) $\xrightarrow{hv:\ 230.5\text{-}462.0\ nm}$ CO+CO+$HO_2$+$HO_2$  IUPAC(2006)

   ③ $C_2H_2O_2$ (Glyoxal) $\xrightarrow{hv:\ 230.5\text{-}462.0\ nm}$ HCHO+CO  IUPAC(2006)

3. $C_4H_6O_2$ (Biacetyl) $\xrightarrow{hv:\ 206\text{-}493\ nm}$ $CH_3CO_3$+$CH_3CO_3$  IUPAC(2011)

4. $C_4H_6O_2$ (Ethylglyoxal) $\xrightarrow{hv:\ 225\text{-}410\ nm}$ $C_2H_5CO_3$ +CO+$HO_2$  IUPAC(2006)

We added the evidence in page 24, lines 546-550 in the modified manuscript:

"The increase in $HO_2$ concentrations afterward was largely attributable to $RO$+$O_2$ reaction/RO decomposition, OH+CO/VOCs reaction, OVOCs photolysis (i.e., $C_3H_4O_2$, $C_2H_2O_2$, $C_4H_6O_2$), and $NO_3$+VOCs reaction, and the increase in $RO_2$ concentrations afterward were largely attributable to OH+VOCs oxidation, OVOCs photolysis and $O_3$+VOCs reaction."

I entertain doubts about the results of the present box model simulation. I think the authors should revalidate the appropriateness of the model thoroughly.

We have re-evaluated the model simulations and added more model simulations to explain some abnormal phenomena in the reference chamber, such as the production and destruction pathways of OH, $HO_2$, $RO_2$, and $NO_3$ during the 3rd-stage 4-min model simulation in the reaction and reference chambers, the $NO_3$+VOC reactions, etc. We found the radicals and gas species in the reaction chamber of the NPOPR detection system were similar to those in the ambient air, while these radicals also unexpectedly existed in the reference chamber. This was mainly because the UV protection film used by the reference chamber did not completely filter out sunlight, which led to the low transmittance of light ranging from 390 nm to 790 nm. We evaluated the interference of the unexpected reactions in the reference chambers and found that the biases of the modeled $P(O_3)_{net}$ caused by this interference using method I and method II were 13.9 % and 22.3 %, respectively; therefore, we noted that ozone photochemical production still existed in the reference chamber, and the measured $P(O_3)_{net}$ by the NPOPR system should be regarded as the lower limit values, as described in page 29, in lines 650-660 in the modified manuscript:

"In conclusion, modeling tests demonstrated that the radicals and gas species in the reaction

chamber of the NPOPR detection system were similar to those in genuine ambient air, while these radicals also unexpectedly existed in the reference chamber. This was mainly because the UV protection film used by the reference chamber did not completely filter out sunlight, which led to the low transmittance of light ranging from 390 nm to 790 nm. The $P(O_3)_{net}$ biases caused by this interference modeled in method I and method II were 13.9 % and 22.3 %, respectively, which ensured that the measured $P(O_3)_{net}$ by the NPOPR detection system should be regarded as the lower limit values of real $P(O_3)_{net}$ in the atmosphere. We recommend that the $J$ values obtained from method I should be used in the model simulation, which can better explain the photochemical formation of $O_3$ in the actual atmosphere, but if direct $J$ value measurements cannot be achieved during field observations, the $J$ values obtained from method II would also be acceptable in modeling studies."

Answers to all of the reviewer's concerns are listed in the following:

- Measurements of NOx using a chemiluminescence NOx monitor.

Did the authors use a commercially available chemiluminescence NOx monitor without further modification? If so, accuracy of $NO_2$ and NOx concentrations would be low because NOz such as $HNO_3$ and PANs interfere observed values of $NO_2$ and NOx. I think this interference could affect discussion in this paper, so the authors should evaluate the interference quantitatively. In the fresh air masses, the interference by descendent pollutants of NOx such as $HNO_3$ and PANs might be small, but that by HONO could be large instead.

Yes, we used a commercially available chemiluminescence NOx monitor without further modification. To investigate the interference of $HNO_3$ and PANs, we added an evaluation of the effect of the $NO_2$ measurement bias of the chemiluminescence NOx monitor on $P(O_3)_{net}$. We compared the $NO_2$ measured by the chemiluminescence NOx monitor with that measured by the CAPS (which is regarded as the trustable $NO_2$ measurement technique without chemical interference) and revealed that a 5% bias could be caused by the chemiluminescence NOx monitor (will be published elsewhere). Therefore, we simulated $P(O_3)_{net}$ by reducing and increasing the mixing ratios of $NO_2$ by 5 % to check the interference caused by using the chemiluminescence NOx monitor to $P(O_3)_{net}$. The results show that increasing and decreasing $NO_2$ by 5 % resulted in a decrease in $P(O_3)_{net}$ by 1.64 % and 3.68 %, respectively. We added this explanation in the modified manuscript in pages 29-30, lines 661-671:"

"Furthermore, because the $NO_2$ data used here were measured by a commercially available chemiluminescence NOx monitor, the $NO_2$ and NOx mixing ratios would be overestimated due to NOz

interference (i.e., $HNO_3$, PANs, HONO, etc.) (Dunlea et al., 2007). According to our test, the chemiluminescence technique could bias $NO_2$ by 5 % compared to the CAPS technique, which is regarded as a trustworthy $NO_2$ measurement technique without chemical interference. Therefore, we simulated the interference of $NO_2$ measured by a chemiluminescence NOx monitor in method I as follows: reducing and increasing the ambient $NO_2$ mixing ratios by 5 % in the 3rd-stage 4-min simulation in the reaction and reference chambers. The results show that increasing and decreasing $NO_2$ by 5 % resulted in a decrease in $P(O_3)_{net}$ by 1.64 % and 3.68 %, respectively (as shown in Fig. S23), which is much smaller than the bias caused by $P(O_3)_{net}$ in the reference chambers (which were 13.9 % and 22.3 % for method I and method II, respectively)."

And added Fig. S23 in the modified supplementary materials:

[Figure]

**Figure S23: $P(O_3)_{net}$ changing in the reaction and reference chambers in method I with ± 5 % of measured $NO_2$.**

Other minor comments:

Some mathematical formulae: The authors should use italic and roman letters correctly.

We have changed the font of the formula to italic and roman letters in the modified manuscript (in lines 56-57, 167, 292, 322, 364, 382, 395-396 and 481-483) and supplementary materials (in line 26, 28, 53-54, 206, 217, 235, 286 and Table S12).

L65: at 424 nm → at < 424 nm

We revised it in lines 65-66 in the modified manuscript: "The specific process of the photochemical reaction is the photolysis of $NO_2$ at $< 420$ nm to generate $O(^3P)$ atoms, thereby promoting the formation of $O_3$ (Sadanaga et al., 2017)."

L72: are proportional to → affect?

We wanted to express that the surface deposition and advection of $O_3$ **are proportional to** ambient $O_3$ mixing ratios, $[O_3]$, which is mainly generated by local photochemistry (Carzorla et al., 2010). We have added the reference in line 73 in the modified manuscript.

Fig. 1: Why do the authors use critical orifices instead of mass flow controllers? Is the temperature of orifices controlled to keep a constant flow rate?

According to our tests, using critical orifices will make the sampling air flow rate in the reaction and reference chambers more stable than using mass flow controllers. We ensured that the air flow rates during the measurement period were constant in the reaction and reference chambers by checking the air flow rates every day during the campaign. Furthermore, the temperature of the orifices did not increase during the sampling time; thus, we assumed that the temperature of the orifices did not affect the air flow rate.

Fig. 1: Are inner walls of the reaction and reference chambers coated with Teflon? If so, please indicate the kind of the Teflon coat.

We did not coat the inner walls of the reaction and reference chambers with Teflon. The reasons are listed as follows: 1) Sadanaga et al. (2017) reported 8–10% dark losses of $O_3$ on uncoated quartz surfaces for a residence time of 21 min in the chambers, which is consistent with the reported dark loss of less than 5% for $O_3$-conditioned flow chambers and a residence time of 4.5 min in Sklaveniti et al. (2018). Sadanaga et al. (2017) indicated that the values of $[NO_2]_{out}$ were in agreement with $[NO_2]_{in}$ within one standard deviation under both dry (0%) and humidified (80%) conditions. The $NO_2$ loss is lower than 5% in both flow chambers and is close to 3% on average in Sklaveniti et al. (2018) under dark conditions. In our study, the wall losses of $NO_2$ were lower than 4% and 2% in the reaction and reference chambers, respectively, and the wall losses of $O_3$ were both lower than 3% in the reaction and reference chambers, as shown in Tables S2 and S3; thus, we assumed this would not cause much bias in our measurement results under dark conditions. 2) Sklaveniti et al. (2018) thought Teflon coating seemed to remove or reduce the photolytic loss of ozone to a negligible level on their instrument because they thought the instrument design reported by Sadanaga et al. (2017) did not seem to be significantly impacted by a photolytic loss of ozone on the quartz flow chambers whose inner surface was coated with Teflon. In

Sadanaga et al. (2017), wall losses of $O_3$ were found to be approximately 10% for both chambers without clear Teflon coating, but wall losses decreased to less than 1.5% when the chambers were coated with Teflon. In our study, we calibrated photolytic $O_3$ losses by performing a set of laboratory and ambient experiments (see sec. S2, pages 18-20 lines 207-237 in the supplementary materials), the results after photolytic $O_3$ loss correction compensated for the photolytic $O_3$ loss interference in the measurement results. 3) We tried to apply Teflon film to the inner walls of the reaction and reference chambers but found that there were some particles produced from the coated wall, which may have been due to bad coating techniques. According to previous studies, particles will take part in the $RO_2/HO_2$ heterogeneous reactions, thus influencing photochemical $O_3$ production. Taking these reasons into consideration, we did not coat the inner walls of the reaction and reference chambers with Teflon; instead, we did the photolytic $O_3$ losses calibration to correct the data, which we think will make our measurement results more accurate.

Table S1: The authors should add standard deviation to average residence times.

Thanks for the suggestions. We have tested in total three sets of experiments at each flow rate, we now added the standard deviation to average residence times in Table S1, we found that for all air flow rates, the standard deviations were <1%, and this may be caused by different operation conditions during the experiments, thus we didn't take this into account when estimate the measured $P(O_3)_{net}$ error, as described in page 17 lines 393-396:

"…the error and LOD of $P(O_3)_{net}$ with a residence time of $\tau$ can be calculated using Eq. (7) and Eq. (8), respectively:

$$P(O_3)_{net\_error} = \frac{\sqrt{(O_{X_\gamma})_{rea\_error}^2 + ((9.72\times[(O_X]_{rea\_measured}^{-1.0024})_{rea\_std})^2 + (O_{X_\gamma})_{ref\_error}^2 + ((9.72\times[(O_X]_{ref\_measured}^{-1.0024})_{ref\_std})^2}}{\tau} \qquad (7)$$

$$LOD = 3\times P(O_3)_{net\_error} \qquad (8)"$$

Fig. S5: The regression lines have non-zero intercepts. These are significant? If so, why? The regression lines for ozone have negative intercepts. In this case, there are large losses of ozone in the high concentration of ozone? For $NO_2$, how about relative humidity in the experiment? Is there no loss of $NO_2$ at high relative humidity?

Thanks for the questions, our answers for the different questions are listed as follows:

(1) The regression lines have non-zero intercepts. These are significant? If so, why? The regression lines for ozone have negative intercepts. In this case, there are large losses of ozone in the high concentration of ozone?

The non-zero intercept is not significant. We added the fittings without an intercept and compared the results with those with an intercept. We found that the wall losses of $O_3$ and $NO_2$ were not much different, and the wall losses affected by the fitting intercepts for $NO_2$ and $O_3$ at an air flow rate of 5 L min$^{-1}$ were all below 4% (as shown in Tables S4 and S5). According to the abovementioned results, we found that when $O_3$ exhibited negative intercepts, the $O_3$ wall losses were still below 4 % at ambient $O_3$ mixing ratios of 0-200 ppb, which was not significant.

We added this explanation in the modified supplementary materials in pages 5-6, lines 73-79:

"The regression lines have non-zero intercepts but not significant. We added the regression fittings without intercept, and compared the regression fitting results with and without intercept (as shown in Figs. S5 and S6). We found that the $O_3$ and $NO_2$ wall losses were not much different (as shown in Tables S2 and S3), and the wall loss affected by the fitting intercepts for $NO_2$ (at ambient mixing ratios of 0-100 ppbv) and $O_3$ (at ambient mixing ratios of 0-200 ppbv) at the air flow rate of 5 L min$^{-1}$ were all below 4 % (as shown in Tables S4 and S5). We found that when the $O_3$ have negative intercepts, the $O_3$ wall losses are still below 4 %, which is not significant."

[Figure]

**Figure S5: Relationship between (a,b) [O₃]ᵢₙ and [O₃]ₒᵤₜ and (c,d) [NO₂]ᵢₙ and [NO₂]ₒᵤₜ in the reaction and reference chambers with intercepts at the flow rates of 1.3, 2, 3, 4, and 5 L min⁻¹, respectively, the solid lines represent the linear fitting of the O₃ or NO₂ mixing ratios at the inlet and outlet of the chambers.**

[Figure]

**Figure S6: Relationship between (a, b) [O₃]ᵢₙ and [O₃]ₒᵤₜ and (c,d) [NO₂]ᵢₙ and [NO₂]ₒᵤₜ in the reaction and reference chambers without intercepts at the flow rates of 1.3, 2, 3, 4, and 5 L min⁻¹, respectively, the solid lines represent the linear fitting of the O₃ or NO₂ mixing ratios at the inlet and outlet of the chambers.**

**Table S2. Wall losses of O₃ and NO₂ of the reaction and reference chambers with intercepts.**

| Flow rate of air (L min⁻¹) | Wall losses of $O_3$ (%) | | Wall losses of $NO_2$ (%) | |
|---|---|---|---|---|
| | Reaction chamber | Reference chamber | Reaction chamber | Reference chamber |
| 1.3 | 2.0 | 2.0 | 4.0 | 2.0 |
| 2 | 0.0 | 1.0 | 4.0 | 1.0 |
| 3 | 0.0 | 1.0 | 2.0 | 0.0 |
| 4 | 0.0 | 0.0 | 0.0 | 0.0 |
| 5 | 0 | 0.7 | 0.3 | 0.6 |

**Table S3. Wall losses of O₃ and NO₂ of the reaction and reference chambers without intercepts.**

| Flow rate of air (L min⁻¹) | Wall losses of O₃ (%) | | Wall losses of NO₂ (%) | |
|---|---|---|---|---|
| | Reaction chamber | Reference chamber | Reaction chamber | Reference chamber |
| 1.3 | 3.0 | 3.0 | 3.0 | 2.0 |
| 2 | 1.0 | 1.0 | 3.0 | 2.0 |
| 3 | 0.0 | 2.0 | 2.0 | 0.0 |
| 4 | 1.0 | 1.0 | 0.0 | 0.0 |
| 5 | 2.0 | 2.0 | 0.0 | 1.0 |

**Table S4. NO₂ wall loss affected by the intercept.**

| Ambient NO₂ mixing ratios (ppbv) | Wall loss affected by the intercept (NO₂, %) | |
|---|---|---|
| | Reaction chamber | Reference chamber |
| 20 | 2.0 | 2.0 |
| 40 | 1.0 | 1.5 |
| 60 | 0.7 | 1.3 |
| 80 | 0.5 | 1.2 |
| 100 | 0.4 | 1.2 |

**Table S5. O₃ wall loss affected by the intercept.**

| Ambient O₃ mixing ratios (ppbv) | Wall loss affected by the intercept (O₃, %) | |
|---|---|---|
| | Reaction chamber | Reference chamber |
| 50 | 3.9 | 2.9 |
| 80 | 2.1 | 2.2 |
| 120 | 1.1 | 1.8 |
| 160 | 0.5 | 1.6 |
| 200 | 0.2 | 1.5 |

(2) For NO₂, how about relative humidity in the experiment? Is there no loss of NO₂ at high relative humidity?

We added this explanation in the modified supplementary materials in page 6,

lines 80-87: "Sklaveniti et al. (2018) found that the wall loss of NO₂ is significantly less than that of O₃ at higher humidity levels. However, in our O₃ photo-enhanced uptake experiments, the wall loss of O₃ was almost unaffected by humidity at a flow rate of 5 L min⁻¹. We also tested the wall losses of NO₂ and O₃ in the chamber at a 5 L min⁻¹ flow rate at different humidities of 35-75 %, the detailed results are shown in Fig. S7 and S8, which shows that the variation in humidity effected the wall loss of NO₂ and O₃ by 0.03-0.12 % and 1.06-1.19 %, respectively, which is much smaller than the instrument detection error (which is 2 % at ambient NO₂ mixing ratios of 0-100 ppb),  thus we didn't count this interference during the data analysis."

[Figure]

**Figure. S7 (a) and (c) represent the NO₂ wall loss at different humidities for the reaction and reference chambers, respectively, (b) and (d) represent the points fitted to all humidities, respectively. Uncertainty in the regression formula was one standard deviation (1σ)."**

L218: low → high?

We retained low. We wanted to say the UV film used in this study still has a low light transmission because it could transmit the lights with wavelength>390 nm.

L226: Why the transmittivity of HONO in the reference chamber is lower than that of $O_3$? How about accuracy and precision of the actinic flux spectrometer?

We measured the transmittivities of all species as follows: we simulated sunlight illumination by adjusting the sunlight (SERIC XG-500B) to provide different intensities of illumination to study the solar UV transmittance through the reaction and reference chambers. The photolysis frequencies of $NO_2$, $O_3$, HONO, etc., inside and outside the reaction and reference chambers were measured using an actinic flux spectrometer (PFS-100; Focused Photonics Inc). The results are shown in Table S7.

The actinic flux spectrometer uses a quartz light receiver head to collect solar radiation from all directions and connects the collected light radiation via optical quartz fibers to the spectrometer, which obtains spectral information in a certain wavelength range and transmits the spectrometer data to the industrial control computer. The computer can convert the signal of the spectral scanning into the actinic flux $F\lambda$ and calculate the light by integrating the actinic flux with the known absorption cross-section $\sigma(\lambda)$ and quantum yield $\varphi(\lambda)$. Therefore, there are no differences when measuring the transmittivities of HONO and $O_3$ on the method and technique aspects. The lower HONO transmittivities in the reference chamber than that of $O_3$ may be due to the UV protection Ultem film on the reference chamber blocking sunlight with wavelengths < 390 nm, as the spectral atlas of HONO was under 190-395 nm at 298 K (IUPAC, 2004), while the spectral atlas of $O_3$ was under 410-750 nm at 298 K (IUPAC, 2004). We added this explanation in the modified manuscript on page 11, lines 254-258:

"The reason for the lower HONO transmittivities in the reference chamber than that of $O_3$ may be that the UV protection Ultem film on the reference chamber blocks sunlight at wavelengths < 390 nm, where the spectral atlas of HONO was under wavelengths of 190-395 nm at 298 K, while that of $O_3$ was under wavelengths of 410-750 nm at 298 K (IUPAC, 2004, http://iupac. pole-ether.fr/)."

We realized that the transmittivities of HONO in the reference chamber in Baier et al. (2015) was also lower than that of $O_3$, which demonstrated that our testing results are reasonable. The actinic flux spectrometer has high resolution and sensitivity for actinic flux measurements. There are no deviations or accuracies in the results of the photochemical flux spectrometer, as there are no standards for reference, but the deviation in spectral resolution is ±0.8 nm for the spectral band range (270-790) nm, which is small.

L228: agree → agreement

We changed "agree" to "agreement" in the modified manuscript in line 246.

Table 1: What is Ultem? There are no definitions in the text.

We added the definition of "Ultem" in modified manuscript in lines 128-130 "... an ultraviolet (UV) protection Ultem film (SH2CLAR, 3 M, Japan) was used to cover the outer surface to block sunlight with wavelengths < 390 nm."

Table 1: The values of 0.019±0.011 should be shaded.

Indeed, we revised it in Table 1.

L272-278 (The airtightness of the reaction and reference chambers): It is hard to follow this section. I think the authors should explain using a schematic diagram for the experiment in the supplement.

Sorry for the confusion description. We have added a schematic diagram for the experiment in the supplementary materials (see Fig. S14), and modified the description accordingly in page13, lines 306-308 in the main manuscript:

"We also checked the airtightness of the reaction and reference chambers by passing through gases with different flow rates based on the schematic diagram shown in Fig. S14

[Figure]

Figure S14. Schematic diagram for investigating the airtightness of the reaction and reference chambers, where MFC1 could measure air flow rate and pressure at the chamber inlet, MFC2 could measure air flow rate and pressure at the chamber outlet."

L297-L300: For calibration of $NO_2$, it is not appropriate to perform calibration of $NO_2$ using a $NO_2$ standard gas because of low reliability. Calibration should be performed using a gas-phase titration method using NO and $O_3$.

We apologize for this mistake. We used $NO_2$ standard gas after we calibrated it using the gas-phase titration method using NO and $O_3$. We used the CAPS $NO_2$ monitor reading as a transition value between the two to obtain the $NO_2$ standard gas and NO+$O_3$ mixing ratios corresponding to the same CAPS $NO_2$ monitor reading. This result showed that the purification of $NO_2$ standard gas is sufficient to calibrate the CAPS $NO_2$ monitor, and we added the related experiments in Fig. S15.

We have added this explanation in the modified supplementary materials in pages 17-18, lines 194-201:

"***Calibration of CAPS NO₂ monitor*** CAPS NO₂ monitor was used to measure the NO₂ standard gas after we have calibrated it using the gas-phase titration method using NO and O₃. We used the CAPS-NO₂ monitor reading as a transition value between the two to obtain the NO₂ standard gas and NO+O₃ mixing ratios corresponding to the same CAPS-NO₂ monitor reading. Results showed the purification of NO₂ standard gas was good enough to calibrate CAPS-NO₂ monitor, as shown in Fig. S15"

[Figure]

**Figure. S15 Correlation between NO₂ standard gas and the NO₂ generated using the gas-phase titration method (NO + O₃).**

Reference: The authors should put the list into alphabetical order.

We have revised it in the modified manuscript and supplementary materials.

**References**

Baier, B. C., Brune, W. H., Lefer, B. L., Miller, D. O., and Martins, D. K.: Direct ozone production rate measurements and their use in assessing ozone source and receptor regions for Houston in 2013, Atmos. Environ., 114, 83-91, http://dx.doi.org/10.1016/j.atmosenv.2015.05.033, 2015.

Cazorla, M., Brune, and W. H.: Measurement of ozone production sensor, Atmos. Meas. Tech., 3, 545-555, https://doi.org/10.5194/amt-3-545-2010, 2010.

Dunlea, E. J., Herndon, S. C., Nelson, D. D., Volkamer, R. M., Martini, F. S., Sheehy, P. M., Zahniser, M. S., Shorter, H., Wormhoudt, J. C., Lamb, B. K., Allwine, E. J., Gaffney, J. S., Marley, N. A., Grutter, M., Marquez, C., Blanco, S., Cardenas, B., Retama, A., Villegas, C. R. R., Kolb, C. E., Molina, L. T., and Molina, M. J.: Evaluation of nitrogen dioxide chemiluminescence monitors in a polluted urban environment, Atmos. Chem. Phys., 7, 2691−2704, https://doi.org/10.5194/acp-7-2691-2007, 2007.

Sadanaga, Y., Kawasaki, S., Tanaka, Y., Kajii, Y., and Bandow, H.: New system for measuring the photochemical ozone production rate in the atmosphere, Environ. Sci. Technol., 51, 2871-2878, https://doi.org/10.1021/acs.est.6b04639, 2017.

Sklaveniti, S., Locoge, N., Stevens, P. S., Wood, E., Kundu, S., and Dusanter, S.: Development of an instrument for direct ozone production rate measurements: measurement reliability and current limitations, Atmos. Meas. Tech., 11, 741-761, https://doi.org/10.5194/amt-11-741-2018, 2018.

Zhou J, Sato K, Bai Y, et al. Kinetics and impacting factors of $HO_2$ uptake onto submicron atmospheric aerosols during the 2019 Air QUAlity Study (AQUAS) in Yokohama, Japan. Atmos. Chem. Phys., 21, 12243-12260, https://doi.org/10.5194/acp-21-12243-2021,2021.

Zhou J, Fukusaki Y, Murano K, et al. Investigation of $HO_2$ uptake mechanisms onto multiple-component ambient aerosols collected in summer and winter time in Yokohama, Japan. J. Environ. Sci., 137, 18-29, https://doi.org/10.1016/j.jes.2023.02.030, 2023.

Kotz, J. C., Treichel, P. M., Townsend, J. and Treichel, D.: Chemistry and Chemical Reactivity. 10th 19 ed., Cengage Learning, 2019.

---

## Author Comment (AC2)

We would like to thank the referee for their recheck and valuable feedbacks, which further improved the quality of the paper. We have addressed the comments point-by-point. The reviewer's comments are in black, our answers are reported in red, and the modifications we made in the manuscript are highlighted in yellow.

**Reviewer #2**

General Comments:

This manuscript details changes made to a two-chamber system used to directly measure ambient net ozone production rates, sensitivity tests conducted on this system, and a final 0-D box modeling comparison of the chemistry in both chambers and in ambient air for a select measurement day in 2021. The criticality of making such direct $P(O_3)_{net}$ measurements in areas with poor air quality for the purpose of determining efficient $O_3$ mitigation strategies is not refuted here; this is an important issue globally. However, while it is clear that much work has gone into the manuscript, it requires major revisions both scientifically and grammatically before being referred for publication. Given the nature of the manuscript, I wonder whether submission to Atmospheric Measurement Techniques is more suitable.

Thanks for the reviewer's insightful comments. As the $P(O_3)_{net}$ measurement system was built based on the dual-channel reaction chamber technique promoted by previous studies, we mainly described how to extend this technique to variable environments by making some improvements and then applied it to a typical $O_3$-polluted area in China to investigate the photochemical $O_3$ formation mechanism. This is the first time we have used the direct $P(O_3)_{net}$ measurement technique in China, and the results extended previous modeling studies and can be used for comparison with our current knowledge. Therefore, we assume that this manuscript is more relevant to the application of such a technique in mechanism exploration than the measurement technique itself; thus, we think its content is relevant to the scope of Atmospheric Chemistry and Physics and will be of interest to its readership. According to the reviewer's suggestions, we have reorganized the manuscript and revised the manuscript scientifically and grammatically.

Specific Comments:
In general, the manuscript is structured appropriately, but has a number of grammatical errors within that prevent clear and fluent presentation. The manuscript would benefit from a review by a technical editor. Some suggestions are presented in the technical corrections below.

Thank you for the reviewer's corrections. we have modified the suggestions point-by-point and then asked a technical editor to review our manuscript to make the description more clearly and fluently.

Equation (1), the ambient $O_3$ budget, is also dependent upon entrainment from the stratosphere. Please add this term for completeness.

Indeed. We added the $O_3$ entrainment from the stratosphere in $O_3$ budget in the modified manuscript in page 3, lines 53-62.

The manuscript indicates that the same tests were conducted on the NPOPR system as in other studies (L188-L189). However, several sensitivity tests conducted in previous studies (Sklaveniti et al., 2018; Baier et al., 2017) are missing from this analysis. These include the following: biases on $P(O_3)_{net}$ from temperature differences between the reaction and reference chamber; short-duration baseline drifting in the CAPS monitor (in presence of both dried and humidified air) and HONO production. Given that some of these additional tests have not been conducted, it is difficult to assess whether the NPOPR system described here has significant improvements over other systems described in the literature. Further, a more detailed description of the system, materials used, switching for sample analysis, etc. would be helpful for characterizing potential differences between this chamber system and others described in previous literature.

Thanks for pointing these out. We have carefully checked the sensitivity tests conducted in previous studies (Sklaveniti et al., 2018; Baier et al., 2017), and conducted some additional tests, the responses for all the points mentioned above are listed as follows:

(1) biases on $P(O_3)_{net}$ from temperature differences between the reaction and reference chamber:

The only difference between the reaction and reference chambers in the NPOPR detection system described here is that the reference chamber was covered by a UV protection Ultem film (SH2CLAR, 3 M, Japan) to block sunlight with wavelengths < 390 nm. This type of film was supposed to not block the heat outside the reference chamber, thus preventing a temperature difference between the two chambers. We measured the temperature both in the reaction and reference chambers when running the NPOPR system in an ambient observation during November 2022 on the Panyu campus of Jinan University in Guangzhou, China (113° 36′ E, 23° 02′ N). We found that the temperature remained the same in both chambers during the measurement period, as shown in Fig. S10.

[Figure]

Figure S10. Air temperature in the reaction and reference chambers during the ambient field observation on Panyu campus of Jinan University.

Accordingly, we added this test in the main manuscript in page 9, lines 200-203:

"We characterized the NPOPR detection system following the same procedures as previous researchers, including the residence time of the air, the wall losses of $NO_2$ and $O_3$, the transmittance of light and temperature differences in the reaction and reference chambers, and the quantitative conversion efficiency of $O_3$ to $NO_2$ ($\alpha$) in the NO-reaction chamber."

and in page 10, lines 234:

"*The light transmittance and temperature differences in the reaction and reference chambers.*"

and in pages 11-12, line 265-270:

"We further detected the temperature in both the reaction and reference chambers when running the NPOPR system in an ambient observation campaign during November 2022 on the Panyu campus of Jinan University in Guangzhou, China (113° 36′ E, 23° 02′ N). We found that the UV protection Ultem film on the reference chamber did not block the heat outside the chamber, and the temperature remained the same in the reaction and reference chambers during the measurement test, as shown in Fig. S10."

(2) short-duration baseline drifting in the CAPS monitor (in presence of both dried and humidified air):

We added the tests for the baseline drift of CAPS at different humidities. As shown in Fig. S11, the CAPS baseline did not shift significantly as the humidify changed:

(a) when injecting ambient air to CAPS, the baseline drifts were 0.035 and 0.032 ppbv (1 $\sigma$) at an integration time of 35 and 100 s, respectively; (b) when injecting wet pure air to CAPS, the baseline drifts were 0.043 and 0.030 ppbv (1 $\sigma$) at an integration time of 35 and 100 s, respectively; (c) when injecting dry pure air to CAPS, the baseline drifts were 0.043 and 0.047 ppbv at an integration time of 35 and 100 s, respectively. Thus, we chose the biggest baseline drift when injecting dry pure air to estimate the $P(O_3)_{net}$ error and calculate the limit of detection of CAPS (which were 0.13 and 0.14 ppbv (3 $\sigma$) at an integration time of 35 and 100 s, respectively, as shown in Sect. 2.3). By doing this, we were able to include all the short-duration baseline drifting in the CAPS monitor under different humidities. Accordingly, we modified the description in page 15, lines 343-357 in the modified manuscript:

"The detailed calibration procedure is as follows: a. injected ~ 10–100 ppbv of NO$_2$ standard gas for 30 min to passivate the surfaces of the monitor and then injected dry pure air for ~ 10 min to minimize the zero point drift, which were 0.043 and 0.047 ppbv at integration times of 35 and 100 s, respectively, and resulted in LODs of CAPS of 0.13 and 0.14 ppbv (3 $\sigma$), respectively; b. injected a wide range of NO$_2$ mixing ratios (from 0–160 ppbv) prepared by mixing the NO$_2$ standard gas with ultrapure air into the CAPS NO$_2$ monitor and repeated the experiments three times at each NO$_2$ mixing ratio. The final results are shown in Fig. 4. To check the baseline drift of the CAPS at different humidities, we added another two sets of tests (as shown in Fig. S11) using ambient air and wet pure air and found that (a) when injecting ambient air into the CAPS (RH ranged from ~30-35%), the baseline drifts were 0.035 and 0.032 ppbv (1 $\sigma$) at integration times of 35 and 100 s, respectively; and (b) when injecting wet pure air into the CAPS (RH ranged from 35-70%), the baseline drifts were 0.043 and 0.030 ppbv (1 $\sigma$) at integration times of 35 and 100 s, respectively. These baseline drifts were smaller than those when injecting dry pure air to estimate the LOD of the CAPS. We chose the largest baseline drift when injecting dry pure air to estimate the $P(O_3)_{net}$ error in the following analysis; by doing this, we were able to include all the short-duration baseline drifting in the CAPS NO$_2$ monitor under different humidities."

[Figure]

**Figure S11. Time series of CAPS baseline and RH when measuring ambient air (a), and when injecting wet (b) and dry (c) pure air in the laboratory, respectively.**

(3) HONO production:

We added the experiments to test the HONO production in the reaction and

reference chambers under environmental conditions similar to those during the SZMGT field observations (humidity of 60-90% at the temperature ~ 20°C and $J$NO$_2$ ~ 0-8 × 10$^{-3}$ s$^{-1}$) at 5 L min$^{-1}$ sampling flow rate. We found that the HONO mixing ratios in the reaction and reference chambers were almost the same and not statistically different with that in the ambient air within the standard deviation, as shown in Fig. S9, therefore, we assume the HONO production in the reaction and reference chambers may not cause a significant difference of $P$(O$_3$)$_{net}$ in the two chambers. Unfortunately, we didn't test HONO during the field observation, but we have added the modeled HONO produced from the precursors before the ambient air was injected into the NPOPR system, as described in Sect. 3.2 (page 22, lines 503-506):

"We used the output mixing ratios of the unmeasured species (i.e., OH, HO$_2$, RO$_2$, SO$_2$, HONO, etc.) from the simulation in the last 1 s of the 2$^{nd}$-stage simulation and all measured values (i.e., O$_3$, NO, NO$_2$, VOCs, $J$ values, RH, $T$, $P$, etc.) as the model input, which were not constrained after providing initial values."

The additional HONO test results were added in the pages 11-12, lines 129-140 in the modified supplementary materials (Fig. S9), the related descriptions are added in pages 14-15, lines 328-337 in the modified manuscript:

"*The HONO production in the reaction and reference chambers* We tested the HONO production in the reaction and reference chambers under weather conditions similar to those during the SZMGT observations (humidity of 60-90% at a temperature of ~ 20 °C and $J$(NO$_2$) of ~ 0-8 × 10$^{-3}$ s$^{-1}$) at a 5 L min$^{-1}$ sampling flow rate. We found that the HONO mixing ratios in the reaction and reference chambers were almost the same and not statistically different from that in the ambient air within the standard deviation, as shown in Fig. S9; therefore, we assumed that the HONO production in the reaction and reference chambers would not cause a significant difference in $P$(O$_3$)$_{net}$ in the two chambers. Unfortunately, we did not test HONO during the field observation period, but we have added the modeled HONO produced from the precursors before the ambient air was injected into the NPOPR system, as described in Sect. 3.2."

[Figure]

Figure. S9 (a)The mixing ratios of HONO in the reaction and reference chambers and (b) the difference of HONO mixing ratios in the reaction and reference chambers.

(4) Given that some of these additional tests have not been conducted, it is difficult to assess whether the NPOPR system described here has significant improvements over other systems described in the literature.

Sorry for missing some of the calibration tests, except all the additional experiments described above, we also described the improvements of the NPOPR system described here over other systems described in the literature in page 30, lines 675-683 in the modified manuscript:

"The main improvements of NPOPR detection system compared to previous studies were as follows: (1) improved the design of the reaction and reference chambers to make sure they have good airtightness; (2) changed the air sampling structure to enable the total air flow rates change freely from 1.3 to 5 L min$^{-1}$ in the reaction and reference chambers, which can make the NPOPR system achieve different limits of detection (LODs) and appliable to different ambient environment; (3) characterized the NPOPR detection system at different air flow rates to optimize the $P(O_3)_{net}$ measurements, the LODs of the NPOPR detection system are 0.07, 1.4, and 2.3 ppbv h$^{-1}$ at air flow rates of 1.3, 3, and 5 L min$^{-1}$, respectively; (4) tested the performance of both reaction and reference chambers by combining the field measurement and the MCM modelling method."

(5) Further, a more detailed description of the system, materials used, switching for sample analysis, etc. would be helpful for characterizing potential differences between this chamber system and others described in previous literature.

The specifications and material of the reaction and reference chambers are described in detail in S1.1. We use 1/2 PFA tubes as the NO-reaction chamber and sampling lines. We used the self-made circuit control software (Four-Channel-

Valves boxed) to control the solenoid valve to realize automatically switch the sampling lines every 2 min. We added the detailed description in page 7 lines 157-159 in the modified manuscript:

"We used homemade circuit control software (Four-Channel-Valves boxed) and a solenoid valve (001-0028-900, Parker, GER) to automatically switch the sampling lines every 2 min."

And page 7 lines 168-175:

"Igor Pro version 6 was used to calculate $P(O_3)_{net}$ as follows: ① separate the data of the reaction and the reference chambers into two sets using the recorded valve number of 1 (reaction chamber) and 0 (reference chamber) during the sampling time; ② for each 2 min period of data, delete the first 20 s and the last 20 s when the signal was not stable, then average the rest data, and do the interpolate calculation of the reference chamber dataset; ③ calculate the difference between the Ox mixing ratios in the reaction and reference chambers (i.e., $\Delta O_X$) at the time when the reaction chamber measured Ox; ④ divide $\Delta O_X$ by the average residence time of air in the reaction chamber $\langle \tau \rangle$ and obtain $P(O_3)_{net}$ at a time resolution of 4 min. "

A full error quantification and discussion of $P(O_3)_{net}$ is missing from the manuscript's main text. Results are presented for several sensitivity tests in the main text, while results from others presented in the supplemental information. However, not all test results that influence the overall $P(O_3)_{net\,\_error}$ are included in the error budget for $P(O_3)_{net}$. This includes the loss of Ox in each chamber, the photo-enhanced Ox loss, residence time uncertainty, etc. A limit of detection is presented in Section 2.4, but is this then later revised in Section 3.2? Each error term in the $P(O_3)_{net\,\_error}$ equation should be clearly defined.

We actually described the error quantification of $P(O_3)_{net}$ when determine the LOD of the NPOPR detection system in Sect. 2.4. To make it clearer, we have changed the title of section 2.4, and reorganized the descriptions, as shown in page 16, line 372 in the modified manuscript:

"**2.4 The measurement error of $P(O_3)_{net}$ and the LOD of the NPOPR detection system**"

And page 16 line 373:

"To assess the measurement error of $P(O_3)_{net}$ and the LOD of the NPOPR detection system…"

And page 16 lines 377-378:

"…as three times the measurement error of $P(O_3)_{net}$, which was determined at a…"

And page 17 lines 393-396:

"…the $P(O_3)_{net\_error}$ and LOD can be calculated using Eq. (7) and Eq. (8), respectively:

$$P(O_3)_{net\_error}=\frac{\sqrt{(O_{X_\gamma})_{rea\_error}^2+((9.72\times[(O_X]_{rea\_measured}^{-1.0024})_{rea\_std})^2+(O_{X_\gamma})_{ref\_error}^2+((9.72\times[(O_X]_{ref\_measured}^{-1.0024})_{ref\_std})^2}}{\tau} \quad (7)$$

$$LOD= 3\times P(O_3)_{net\_error} \quad (8)$$"

And page 17 lines 402-404:

"In conclusion, the LOD of the NPOPR detection system is determined to be three times $P(O_3)_{net\_error}$, where $P(O_3)_{net\_error}$ is mainly determined by the measurement error of Ox (including the Ox measurement error of the CAPS $NO_2$ monitor, the light-enhanced loss of $O_3$, and the chamber Ox losses)."

From the description above, we have included all the test results that influence the overall $P(O_3)_{net\_error}$, and defined them accordingly, which included the Ox measurement error of the CAPS $NO_2$ monitor, the loss of Ox in each chamber, and the photo-enhanced Ox loss, residence time uncertainty, etc. We haven't change the LOD later in Sect. 3.2, instead, we have corrected the measured data according to the errors caused by these interferences to compare them with the modeled data.

The potential for Ox loss at high RH and photo-enhanced Ox loss was discussed in supplemental information and a correction factor was devised to exclude this bias, but it is unclear how this correction factor is (or can be) used for ambient measurements; b) which RH range (30-32% is shown, but ambient RH routinely exceeds this value) was tested; c) for which flow rate regime was this correction devised and d) whether this relationship holds for all measurement scenarios. More details are needed here to fully understand the test being conducted and how it can be applied to correct $P(O_3)_{net}$.

Sorry for the unclear description. We have added the light-enhanced $O_3$ loss quantification method in the modified supplementary materials in page 19, lines 226-228:

"When quantifying the light-enhanced $O_3$ loss ($d[O_3]$) during ambient air measurement, we first calculate $\gamma$ using the measured $J(O^1D)$ and the $\gamma$ -$J(O^1D)$ equations listed in Fig. S16a in the reaction and reference chambers, then using the measured $[O_3]$ and Eq. S6 to calculate $d[O_3]$."

The answers for the following questions are listed as follows:

b) which RH range (30-32% is shown, but ambient RH routinely exceeds this value) was tested;

We tested a wide range of RH, and we have added the $O_3$ loss experiments with humidities of 35, 50 and 75 % in Fig. S16a. We found that the $\gamma$ -$J(O^1D)$ equation obtained here is also suitable for higher RH conditions. Together with the light-enhanced loss of $O_3$ as a function of RH shown in Fig. S16b, we believe that RH will not further affect the light-enhanced loss of $O_3$.

c) for which flow rate regime was this correction devised and d) whether this relationship holds for all measurement scenarios. More details are needed here to fully understand the test being conducted and how it can be applied to correct $P(O_3)_{net}$.

c) and d) The flow rate was set at 5 L min$^{-1}$ when performing this test, which is the same as the flow rates used in the field observations. However, we also tested this relationship at other flow rates (i.e., at a flow rate of 2 L min$^{-1}$) and found that this relationship was different at different flow rates, which means that we have to perform such calibration every time we use a different flow rate. We have added this statement on page 18, lines 207-208 in the supplementary materials:

"*The light-enhanced loss of O₃ in the reaction and reference chambers* at 5 L min$^{-1}$ (the ambient observation used flow rate in this study) were investigated by carrying out the following experiments: …"

It is unclear what new information is being presented with the discussion of flow state in L282-L290 and I might suggest that this section could be removed. While the flow may be considered laminar, it is shown from residence time testing that the fluid within the chambers does not represent a "plug" flow, so reactions on the chamber walls may still produce biases in $P(O_3)_{net}$.

Thank you for the suggestion. We think the discussion on L282-L290 give us reasonable support for checking the flow states at different flow rates in the reaction and reference chambers; thus, we prefer to keep it like this. The fluid showed a peak at the outlet of the chamber when performing the residence time testing because we only injected high concentrations of $NO_2$ for 20 s, and its concentration showed an increase at first and then a decreasing trend due to the dilution effect. If it is a "plug" flow, the theoretical time is equal to the volume divided by the air flow rate, and the average residence time we measured was close to the theoretical time, as shown in Table S1.

Units of ppbv imply a "mixing ratio" in atmospheric chemistry terms, whereas a "concentration" is often referred to in molec cm$^{-3}$ or mol L$^{-1}$.

Indeed, we replaced "ppbv" with "mixing ratio" throughout the manuscript.

Have the authors investigated the potential for a small amount of $NO_2$ impurity in the NO mixture used for conversion and can you address this potential bias in $P(O_3)_{net}$?

During the experiments, to avoid the influence of the small amount of $NO_2$ impurity in the NO mixture used for conversion, we added a cylinder filled with partialized crystals of $FeSO_4 \cdot 7H_2O$ to reduce $NO_2$ in the $NO/N_2$ gas cylinder to NO. We modified the related description in the modified manuscript in page 12, lines 275-277:

"To avoid the influence of small amounts of $NO_2$ impurity in the NO standard gas used for conversion, we added a cylinder filled with partialized crystals of $FeSO_4 \cdot 7H_2O$ to reduce $NO_2$ in the $NO/N_2$ gas cylinder to NO…"

To test the capacity of the particulate crystals of $FeSO_4 \cdot 7H_2O$ to reduce $NO_2$, we injected ~1800 ppbv NO into the NO-reaction chamber, and tested the $NO_2$ mixing ratios from the outlet of it using CAPS-$NO_2$ monitor, the results are shown in Fig. S13. We found that the standard deviation of $NO_2$ mixing ratios was lower than 0.05 ppbv, which is smaller than the precision of CAPS, so we believe the particulate crystals of $FeSO_4 \cdot 7H_2O$ performs well and the potential bias introduced by NO for $P(O_3)_{net}$ is negligible. We modified the related description in the modified manuscript in page 12, lines 277-283:

"We injected ~1800 ppbv NO into the NO-reaction chamber and tested the $NO_2$ mixing ratios from its outlet using a CAPS $NO_2$ monitor, as shown in Fig. S13. We found that the standard deviation of the $NO_2$ mixing ratios was lower than 0.027 ppbv, which is smaller than the baseline drifts of the CAPS (which were 0.043 and 0.030 ppbv (1 $\sigma$) at integration times of 35 and 100 s, respectively, as mentioned in Sect. 2.3), so we believe the particulate crystals of $FeSO_4 \cdot 7H_2O$ performed well and the potential bias introduced by the impurity in NO mixing ratio for $P(O_3)_{net}$ was negligible."

[Figure]

Figure. S13 Time series of NO$_2$ when injecting NO into CAPS-NO$_2$ monitor.

L330-339 addresses the fact that Ox (= NO$_2$+O$_3$) mixing ratios change over the 2 minutes sampling time of each chamber. How do the authors address issues of NO$_2$ atmospheric variability over the sampling time of the reaction versus the reference chamber, and the subtraction of these alternating two measurements, especially in urban areas? It is difficult to understand the resolution of $P$(O$_3$)$_{net}$ in Figure 6 and how the data are averaged (or not?). Can you provide more details on how the data are processed to help the discussion?

Ox mixing ratios do slightly change over the 2 min sampling time of each chamber. To address the issues of NO$_2$ atmospheric variability over the sampling time of the reaction versus the reference chamber, we used the interpolated Ox data in the reference chamber to make the Ox background of the reaction chamber closer to the real condition. More details are added in the modified manuscript in page7, lines 168-175:

"Igor Pro version 6 was used to calculate $P$(O$_3$)$_{net}$ as follows: ① separate the data of the reaction and the reference chambers into two sets using the recorded valve number of 1 (reaction chamber) and 0 (reference chamber) during the sampling time; ② for each 2 min period of data, delete the first 20 s and the last 20 s when the signal was not stable, then average the rest data, and do the interpolate calculation of the reference chamber dataset; ③ calculate the difference between the Ox mixing ratios in the reaction and reference chambers (i.e., $\Delta$O$_X$) at the time when the reaction chamber measured Ox; ④ divide $\Delta$O$_X$ by the average residence time of air in the reaction chamber $\langle\tau\rangle$ and obtain $P$(O$_3$)$_{net}$ at a time resolution of 4 min."

The data sets used in Fig. 6 have with a time resolution of 1 h, averaged from the

obtained 4 min time resolution $P(O_3)_{net}$ as described above, so as to facilitate comparison with other data.

The modeling section could use some work to clarify and condense the information presented. In theory, modeling of the chemistry in both chambers separately seems like a good check on what is actually measured by the NPOPR system, but it is known from previous studies (and this issue is presented here as well) that modeling of $HO_2$ and $RO_2$ in ambient air does not match that which is measured (e.g. from Ren et al. 2013). Thus, if $HO_2$ and $RO_2$ are not well-captured in the model from parameterized VOCs and reactions therein, it is difficult to use the model to verify the chamber chemistry. Therefore, one suggestion could be to simplify this discussion to compare the ambient photochemistry from the model to the NPOPR system, given modeling limitations.

Indeed, if $HO_2$ and $RO_2$ are not well captured in the model from parameterized VOCs and reactions therein, it is difficult to use the model to verify the chamber chemistry. To evaluate $P(O_3)_{net}$ in the ambient air, we added another model, which maintains the setup conditions of the 2nd-stage during the 3rd-stage 4-min simulation. We added the obtained $P(O_3)_{net}$ value in the ambient air at 12:04 on 7 December to Figs. 10 and S22, which was 24.63 ppbv h$^{-1}$, 1.4 ppbv h$^{-1}$ lower than the measured value. Therefore, we believe that the results of this simulation are more accurate and that the analysis of the simulations in the chambers can help us to assess the ozone photochemical production mechanism in the chamber.

And we added the description in page 22, lines 506-509 in the modified manuscript:

"In addition, while maintaining the setup conditions for the 2nd-stage of the simulation, we extended the simulation of the environment to 12:04 to obtain the modeled $P(O_3)_{net}$ in the environment in the 3rd-stage simulation . The result is shown in orange marker in Fig. 10d."

and page 28, lines 627-629 in the modified manuscript:

"…which was 1.4 ppbv h$^{-1}$ higher than the modeled $P(O_3)_{net}$ value in the ambient air (orange marker in Fig. 10d, 24.6 ppbv h$^{-1}$)…"

Technical Corrections:
- L87: You may also choose to reference Baier et al. (2017) here for similar work.

We tested wall losses of $NO_2$ and $O_3$ in the chamber at a 5 L min$^{-1}$ flow rate at different humidities of 35-75 %, as well as HONO mixing ratios in the reaction and reference chambers under weather conditions similar to those during the Shenzhen observations (humidity of 60-90% while the temperature was approximately 20 °C and $J(NO_2)$ of 0-8 × 10$^{-3}$ s$^{-1}$), and we added the description

on page 10, lines 229-232 in the modified manuscript:

"We also tested the wall losses of $NO_2$ and $O_3$ in the chamber at a 5 L $min^{-1}$ flow rate at different humidities of 35-75 %, the detailed results are shown in Fig. S7 and S8, which shows that the variation in humidity effected the wall loss of $NO_2$ and $O_3$ by 0.03-0.12 % and 1.06-1.19 %, respectively, which is much smaller than the instrument detection error (which is 2 % at ambient $NO_2$ mixing ratios of 0-100 ppb) …"

And we added the results in pages 10-11, lines 119-128 in the modified supplementary materials.

And the additional HONO test results were added in the pages 11-12, lines 129-140 in the modified supplementary materials (Fig. S9), the related descriptions are added in pages 14-15, lines 328-337 in the modified manuscript:

"***The HONO production in the reaction and reference chambers*** We tested the HONO production in the reaction and reference chambers under weather conditions similar to those during the SZMGT observations (humidity of 60-90% at a temperature of ~ 20 °C and $J(NO_2)$ of ~ 0-8 × $10^{-3}$ $s^{-1}$) at a 5 L $min^{-1}$ sampling flow rate. We found that the HONO mixing ratios in the reaction and reference chambers were almost the same and not statistically different from that in the ambient air within the standard deviation, as shown in Fig. S9; therefore, we assumed that the HONO production in the reaction and reference chambers would not cause a significant difference in $P(O_3)_{net}$ in the two chambers. Unfortunately, we did not test HONO during the field observation period, but we have added the modeled HONO produced from the precursors before the ambient air was injected into the NPOPR system, as described in Sect. 3.2."

- L90: There is no Baier et al. (2021) in references. Do you mean Baier et al. (2015)?

We apologize for this mistake. We meant Baier (2015), we revised it in lines 90-92 in the modified manuscript:

"Recently, researchers have developed sensors that can directly measure $P(O_3)_{net}$ in the atmosphere using the dual-channel chamber technique (Sadanaga et al., 2017; Cazorla et al., 2010; Baier et al., 2015 and 2017; Sklaveniti et al., 2018)…"

"…the sensors developed by Cazorla et al. (2010) and Baier et al. (2015) both have an $NO_2$-to-$O_3$ converter unit…"

- L98: Could also benefit from citing Baier et al., 2017 for chamber artifact discussion

Yes, we have cited Baier et al. (2017) for a discussion chamber artifact in Sec. 1.3 and Sec. 1.4 in the supplementary materials. We tested the $NO_2$ wall loss under different humidities and mixing ratios of HONO in the reaction and reference chambers as mentioned in Baier et al., (2017), and added the related discussion in the modified manuscript and supplementary materials as mentioned above.

- L107-108: please re-write for clarity

We have revised it in lines 108-109 in the modified manuscript:

"Furthermore, all the current sensors have different degrees of wall loss of $NO_2$ and $O_3$ that can even reach 15 %, which largely affect the accuracy of the evaluation of $P(O_3)_{net}$."

- L116: change "access" to assess

We changed it in line 117 in the modified manuscript:

"…which allowed us to assess the ability of the current modeling method to model $P(O_3)_{net}$, as described in Sect. 3."

- L131: change "amounted" to mounted

We changed it in line 132:

"…a Teflon filter was mounted before the chamber inlet to remove fine particles"

- L132: Please describe alternating flow more clearly. How do your account for the "transition" period after switching and sampling by the CAPS monitor (i.e. is some portion of the data discarded after transitioning between reaction and reference chambers as in Sklaveniti et al., 2018?)

We used CAPS $NO_2$ monitor to alternately detect the gas from the reaction and the reference chambers through the Teflon valve shown in Fig. 1. There was a transition period of about 20 s after each valve cutting, and we discarded the data

of 20 s after transitioning between reaction and reference chambers as in Sklaveniti et al., (2018), more details can be found in page 7, lines 168-175 as mentioned above.

- L141: How do you assess portability?

It consists of CAPS-$NO_2$ monitor, indoor cabinets to put the CAPS-$NO_2$ monitor, the automatic sampling system, and the automatic data sampling system, outdoor dual chambers with the push-pull base. We have assembled each part together to make it easy to transport (as shown in Fig. 1b). We added the related description in page 6, lines139-145 in the modified manuscript as follows:

"Compared to previous studies that used a dual-channel UV-absorption $O_3$ monitor (Cazorla et al., 2010) or the LIF-$NO_2$ monitor (Sadanaga et al., 2017) for Ox measurements, our choice could make the NPOPR detection system have a more stable zero-baseline and be more portable by assembling each part together, i.e., put the CAPS $NO_2$ monitor, the automatic sampling system, and the automatic data sampling system onto the indoor cabinets with the push-pull base, and put the dual chambers onto the outdoor shelf with the push-pull base."

- L186: Accuracy implies that $P(O_3)$ is produced as it is in the atmosphere, which cannot be determined here. Better phrasing could be: "to make the NPOPR less prone to biases than other systems" or similar.

We have revised it in page 9, lines 197-198: "These efforts made the NPOPR system less prone to biases than other systems and increased its applicability."

- L217: Please calculate the bias in $P(O_3)_{net}$ incurred from Ox loss in the reaction and reference chamber.

We actually calculate the bias in $P(O_3)_{net}$ incurred from Ox loss in the reaction and reference chamber when determine the LOD of the NPOPR detection system in Sect. 2.4. To make it clearer, we have changed the title of section 2.4, and reorganized the descriptions, as shown in page 16, line 372 in the modified manuscript:

"**2.4 The measurement error of $P(O_3)_{net}$ and the LOD of the NPOPR detection system**"

And page 16 lines 373:

"To assess the measurement error of $P(O_3)_{net}$ and the LOD of the NPOPR detection system…"

"…as three times the measurement error of $P(O_3)_{net}$, which was determined at a…"

"…the error and LOD of $P(O_3)_{net}$ with a residence time of $\tau$ can be calculated using Eq. (7) and Eq. (8), respectively:

$$P(O_3)_{net\_error}=\frac{\sqrt{(O_X\gamma)_{rea\_error}^2+((9.72\times[(O_X]_{rea\_measured}^{-1.0024})_{rea\_std})^2 +(O_X\gamma)_{ref\_error}^2+((9.72\times[(O_X]_{ref\_measured}^{-1.0024})_{ref\_std})^2}}{\tau} \quad (7)$$

$$LOD= 3\times P(O_3)_{net\_error} \quad (8)"$$

"In conclusion, the LOD of the NPOPR detection system is determined to be three times $P(O_3)_{net\_error}$, where $P(O_3)_{net\_error}$ is mainly determined by the measurement error of Ox (including the Ox measurement error of the CAPS $NO_2$ monitor, the light-enhanced loss of $O_3$, and the chamber Ox losses)."

From the description above, we have included all the test results that influence the overall $P(O_3)_{net\_error}$, and defined them accordingly, which included the Ox measurement error of the CAPS $NO_2$ monitor, the loss of Ox in each chamber, the photo-enhanced Ox loss, residence time uncertainty, etc.

- Table 1: is some shading missing here?

Indeed, we added shadows to 0.019±0.011 in Table 1.

- Figure 4: Typically, the axes are reversed for a calibration such that y represents known values and x represents those values that are calibrated.

Thank you for your advice. Here we make x represents known values and y represents those values that are measured by CAPS-$NO_2$ monitor, because we wanted to use this equation to access the measurement error of CAPS-$NO_2$ monitor, thus benefit us to estimate the measurement error of $P(O_3)_{net}$, which does not affect the analysis results of the experimental data.

- L360: Please describe what 'cd' means for readers

We have described it in line 418 in the manuscript: "…where cd indicates the light intensity

SI unit candela."

- Figure 5: Please indicate shading in a color bar, etc.

The color bar indicates the light intensity change, the lighter the color, the less light intensity. We removed the color bar in the modified manuscript as we think the color bar here is unnecessary.

- Figure 6b): is this an average of the diurnal $P(O_3)$

Yes. Figure 6b) shows the average diurnal variation of different parameters (including $T$, RH, $J_{NO2}$, $J_{O1D}$, NO, $NO_2$, $NO_X$, $O_X$, $O_3$, $P(O_3)_{net}$) during 7 to 9 December 2021. We have changed the description of Fig. 6 as follows in page19, lines 441-444 in the modified manuscript:

"**Figure 6: (a) Time series and (b) average diurnal variations of $P(O_3)_{net}$, $J(NO_2)$, $J(O^1D)$, T, RH, $O_X$, $NO_2$ and NO measured at SZMGT from 7 to 9 December 2021. The shaded areas represent the error of each measured species, where the error of $P(O_3)_{net}$ was calculated according to the method described in Appendix II (the estimation of the $P(O_3)_{net}$ error).**"

- L432: November or December?

December. We have revised it in line 493 in the modified manuscript.

- SI L129: perhaps change the word "ingestion" to "loss?

We agree with the reviewer's suggestion. We have changed "ingestion" to "loss" in lines 366, 390, 397 in the modified manuscript, and lines 193, 207, 213, 215, 216, 229 in the modified supplementary materials.

- SI L135-136 should be Fig. S7a, not S6a

Thank you. We have checked the figure sequence again after the modifications.

- Figure S7: please add units on figure axes

Ok. We added the units in the modified supplementary materials in Fig. S16.

- Table S7a: Sklaveniti et al., 2018 does not constitute an urban area – please see description within this particular reference and check other sites as well.

Sorry for the confusion description. We have checked the observation location of Sklaveniti (2018) was at a site 2.5 km northeast of the Indiana University Bloomington campus, we have revised it in Table S10a.

- SI Tables S7a/S7b require corrections made to locations. For example, Writtle College is not located in the USA; Houston, USA should be replaced with City, State, Country format like other sites: Houston, Texas, USA, etc.

We have checked the sites and marked the revisions in yellow in Tables S10a/S10b.

Appendix:

We have detected more errors and modified them in the manuscript, which are listed as follows:

1. We have added a co-author, Yaqing Zhou, for her help in the experiment of the HONO production.
2. We changed "minutes" to "min" in modified manuscript and the supplementary materials.
3. We changed "hours" to "h" in modified manuscript in line 374.
4. We revised the subscript of the Eq. (7) in in modified manuscript in line 395.

**References**

Baier, B. C., Brune, W. H., Lefer, B. L., Miller, D. O., and Martins, D. K.: Direct ozone production rate measurements and their use in assessing ozone source and receptor regions for Houston in 2013, Atmos. Environ., 114, 83-91, http://dx.doi.org/10.1016/j.atmosenv.2015.05.033, 2015.

Baier B C, Brune W H, Miller D O, et al., Higher measured than modeled ozone production at increased NOx levels in the Colorado Front Range, Atmos. Chem. Phys., 17: 11273–11292, https://doi.org/10.5194/acp-17-11273-2017, 2017.

Cazorla, M., Brune, and W. H.: Measurement of ozone production sensor, Atmos. Meas. Tech., 3, 545-555, https://doi.org/10.5194/amt-3-545-2010, 2010.

Ren, X., van Duin, D., Cazorla, M., Chen, S., Mao, J., Zhang, L., Brune, W. H., Flynn, J. H., Grossberg, N., Lefer, B. L., Rappenglück, B., Wong, K. W., Tsai, C., Stutz, J., Dibb, J. E., Thomas Jobson, B., Luke, W. T., and Kelley, P.: Atmospheric oxidation chemistry and ozone production: results from SHARP 2009 in Houston, Texas, J. Geophys. Res.-Atmos., 118, 5770-5780, https://doi.org/10.1002/jgrd.50342, 2013.

Sadanaga, Y., Kawasaki, S., Tanaka, Y., Kajii, Y., and Bandow, H.: New system for measuring the photochemical ozone production rate in the atmosphere, Environ. Sci. Technol., 51, 2871-2878, https://doi.org/10.1021/acs.est.6b04639, 2017.

Sklaveniti, S., Locoge, N., Stevens, P. S., Wood, E., Kundu, S., and Dusanter, S.: Development of an instrument for direct ozone production rate measurements: measurement reliability and current limitations, Atmos. Meas. Tech., 11, 741-761, https://doi.org/10.5194/amt-11-741-2018, 2018.

---

## Author Response (AR2)

We thank the reviewer's further comments and suggestions, which will assist us in providing a more accurate description of our work. Our answers are listed in the following in red, after the referee's comments, which are in black. The modifications in the text are marked in yellow.

**Editor:**

The authors have addressed my initial major concern. I recommend this manuscript to be published after the authors revise a minor point.

It remains unconvinced why the transmittivity of HONO in the reference chamber is lower than that of $O_3$ even if I read your response.

According to JPL Publication 19-5, absorption cross section of ozone and quantum yield to form O(1D) at wavelengths of 390-410 is about $1 \times 10^{-23}$ $cm^2$ and 0.08, respectively. On the other hand, Absorption cross section of HONO and photolysis quantum yield at wavelengths of 390-395 is the order of $10^{-21}$ $cm^2$ (two or three order of magnitude higher than that of ozone) and unity (about ten times higher than that of ozone), respectively.

The authors state "the spectral atlas of HONO was under wavelengths of 190-395 nm at 298 K, while that of $O_3$ was under wavelengths of 410-750 nm at 298 K". But the quantum yield to form O(1D) is 0 at wavelengths longer than 411 nm. The threshold wavelength of the photodissociation to form O(1D) is 411 nm in terms of binding energy of ozone.

In summary, I cannot believe the transmittivity of HONO in the reference chamber is lower than that of $O_3$.

**Questions/Suggestions for improvement:**

The authors should re-calculate $J$ values and/or re-evaluate measurement errors.

The issue mentioned by reviewer is critical, sorry for the unclear description. The reviewer is correct that according to JPL Publication 19-5, absorption cross section of HONO at wavelengths of 390-395 ranged from $\sim 4.0\text{-}17.1 \times 10^{-21}$ $cm^2$, which is about two or three orders of magnitude higher than that of ozone (ranged from $\sim 0.8\text{-}2.6 \times 10^{-23}$ $cm^2$ at wavelengths of 390-410 nm), and the photolysis quantum yield of HONO at wavelengths of 390-395 is unity, which is about ten times higher than that of ozone ($\sim 0.08$). This will surely make the $J$ values of HONO inside the reference chamber (which only has sunlight with wavelengths > 390 nm) higher than that of ozone, according to the following equation:

$$J_{\text{value TUV}} = \int_a^b \delta_i \times \phi_i \times F_i \, \Delta\lambda_i$$

where $a$ and $b$ represent the range of the set wavelength, $\delta_i$, $\phi_i$, and $F_i$ stand for the absorption cross section, quantum yield, and spectral actinic flux of the species $i$, respectively.

We also find that the transmittivity of HONO and $O_3$ in the reference chamber,

obtained from the TUV simulation as described in Sect. 3.2, were 0.01 and 0, respectively, which are shown in Table S13. Therefore, we believe the non-zero measurement results of the transmittivity of $O_3$ shown in Table 1 and Table S7 are mostly probably due to the instrument measurement error, this error is relatively large due to a limited number of measurement points (3 points for each species).

According to the working theory of the actinic flux spectrometer, the measurement error may rise from the angular response deviation of the quartz receiver head. According to Bohn et al. (2017), the measurement error of the actinic flux spectrometer can reach $\pm 5$ %. According to this, we re-evaluated the transmittivity error listed in Table 1 and Table S7 as follows: ① calculate the absolute measurement error of all measured $J$ values inside and outside the reaction and reference chambers based on the $\pm 5$ % instrument measurement error; ② calculate the average values of all the measured $J$ values (including ($J(NO_2)$, $J(O1D)$, $J(HONO)$, $J(H_2O_2)$, $J(NO_3\_M)$, $J(NO_3\_R)$, $J(HCHO\_M)$, and $J(HCHO\_R)$) inside and outside the chambers; ③calculate the propagated error of transmittivity, using the following error propagation equation:

$$\sigma_{Transmittivity} = \sqrt{(\frac{\sigma_{J\,value\,in}}{A_{J\,value\,in}})^2 + (\frac{\sigma_{J\,value\,out}}{A_{J\,value\,out}})^2}$$

where $\sigma_{Transmittivity}$ represents the transmittivity error; $\sigma_{J\,value\,in}$ and $\sigma_{J\,value\,out}$ represent the measurement error of $J$ value inside and outside the chambers, respectively; $A_{J\,value\,in}$ and $A_{J\,value\,out}$ represent the average $J$ values measured inside and outside the chambers, respectively.

We find that the calculated transmittivity errors are 0.07 for all $J$ values. Within this error range, the $J(O1D)$, $J(HONO)$, $J(H_2O_2)$, $J(HCHO\_M)$, and $J(HCHO\_R)$ can be considered statistically indistinguishable from 0 in the reference chamber, however, $J(NO_2)$, $J(NO_3\_M)$, and $J(NO_3\_R)$ still exhibit positive values.

To evaluate $P(O_3)_{net}$ error caused by the measurement error of $J$ values, we introduced a $\pm 5$ % error to the measured $J$ values during the 3rd stage of the 4-min simulation in method I, the modeled $P(O_3)_{net}$ results are added in Fig. S24 in the modified supplementary. We found that the inclusion of a -5 % $J$ values measurement error can lead to a decrease in $P(O_3)_{net}$ by 7.27 %, while adding a +5 % $J$ values measurement error can cause an increase in $P(O_3)_{net}$ by 3.08 %. This implies that the maximum bias of $P(O_3)_{net}$ caused by the measurement error of $J$ values falls within the error range of the currently assessed $P(O_3)_{net}$ error, which was 13.9 % for method I. Therefore, we conclude that the transmittivity of HONO in the reference chamber is not statistically different from that of $O_3$ within the measurement error of $J$ values, and this type of error will not influence our final modeling results and conclusions.

We have included the aforementioned analyses and revised the previous

inaccurate descriptions in the revised manuscript on page 11, lines 246-270:

"The photolysis frequencies of all species inside the reaction chamber were in agreement with those measured outside the reaction chamber within 4 %. Table S7 shows that the transmittivities of $J(H_2O_2)$, $J(NO_3\_M)$, $J(NO_3\_R)$, $J(HCHO\_M)$, and $J(HCHO\_R)$ in the reaction chamber were more than 90 %. However, we have observed that the transmittivities of $J(O^1D)$ were even higher than those of $J(HONO)$ (as shown in Table 1) in the reference chamber (which blocks sunlight at wavelengths < 390 nm), theoretically, this is not possible according to JPL Publication 19-5 (Burkholder et al., 2020), where the absorption cross section of HONO at wavelengths of 390-395 ranged from approximately $4.0\text{-}17.1\times10^{-21}$ $cm^2$, which is about two or three orders of magnitude higher than that of ozone (which ranged from approximately $0.8\text{-}2.6\times10^{-23}$ $cm^2$ at wavelengths of 390-410 nm), and the photolysis quantum yield of HONO at wavelengths of 390-395 is unity, which is about ten times higher than that of ozone (~ 0.08). This will surely make the $J$ values of HONO inside the reference chamber (which only has sunlight with wavelengths > 390 nm) higher than that of ozone, according to the Eq. (S9). We also found that the transmittivity of HONO and $O_3$ in the reference chamber obtained from the TUV simulation (as described in Sect. 3.2) were 0.01 and 0, respectively, as shown in Table S13. Therefore, we believe the non-zero measurement results of the transmittivity of $O_3$ shown in Table 1 and Table S7 are mostly probably due to the instrument measurement error, this error is relatively large due to a limit number of measurement points (3 points for each species). We further evaluated the measurement error of $J$ values based on the instrument measurement error of the actinic flux spectrometer, which can reach $\pm 5$ % according to Bohn et al. (2017), and re-evaluated the transmittivity error listed in Table 1 and Table S7 following the procedures described in supplementary materials (Sect. 1.5). The calculation result from Eq. (S5) show that the transmittivities errors are 0.07 for all species, within this error range, $J(O1D)$, $J(HONO)$, $J(H_2O_2)$, $J(HCHO\_M)$, and $J(HCHO\_R)$ can be considered statistically indistinguishable from 0 in reference chamber. However, $J(NO_2)$, $J(NO_3\_M)$, and $J(NO_3\_R)$ still distinctly positive values. Specifically, the transmittivities of $J(NO_3\_M)$ and $J(NO_3\_R)$ of the reference chamber were more than 90 % (Table S7). The influence of the measurement error of $J$ values of all species on $P(O_3)_{net}$ will be discussed in Sect. 3. "

And added the additional explanation in the modified supplementary on pages 12, lines 151-163:

"According to the working theory of the actinic flux spectrometer, the measurement error may rise from the angular response deviation of the quartz receiver head. According to Bohn et al. (2017), the measurement error of the actinic flux spectrometer can reach±5 %. According to this, we re-evaluated the transmittivity error listed in Table 1 and Table S7 as follows:①calculate the absolute measurement error of all measured $J$ values inside and outside the reaction and reference chambers based on the ±5 % instrument measurement error; ②calculate the average values of all the measured $J$ values (including ($J(NO_2)$, $J(O1D)$, $J(HONO)$, $J(H_2O_2)$, $J(NO_3\_M)$, $J(NO_3\_R)$, $J(HCHO\_M)$, and $J(HCHO\_R)$)) inside and outside the chambers; ③calculate the propagated error of transmittivity, using the following error propagation equation:

$$\sigma'_{\text{Transmittivity}}= \sqrt{\left(\frac{\sigma'_{J \text{ value in}}}{A_{J \text{ value in}}}\right)^2 + \left(\frac{\sigma'_{J \text{ value out}}}{A_{J \text{ value out}}}\right)^2} \tag{S5}$$

where $\sigma'_{\text{Transmittivity}}$ represents the transmittivity error; $\sigma'_{J \text{ value in}}$ and $\sigma'_{J \text{ value out}}$ represent the measurement error of $J$ value inside and outside the chambers, respectively; $A_{J \text{ value in}}$ and $A_{J \text{ value out}}$ represent the average $J$ values measured inside and outside the chambers, respectively."

And modified the caption of Table1 in the modified manuscript accordingly on page 12, lines 278-285:

"Table 1. Transmittivities of photolysis frequency $J$ ($s^{-1}$) values of different species in the reaction and reference chambers. The shaded and clear regions correspond to the transmittivities of $J$ values in the reference (Ultem coated) and reaction (clear) chambers, respectively. The "transmittivities" column shows the transmittivities of the tested species from the measurements conducted with the set photolysis frequencies using SERIC XG-500B sunlight (this study) and ambient (literature). It should be noted that the errors listed here are relatively large and may not reliable due to a limit number of measurement points (3 points for each species). The calculated transmittivity errors are 0.07 for all species based on the ±5 % measurement error of the instrument."

As well as the caption of Table S7 in the modified supplementary materials on page 13, lines 165-172:

"Table S7. Photolysis frequency $J$ ($s^{-1}$) of different species and the transmittivities of $J$ values in the reaction and reference chambers. The shaded and clear regions correspond to the photolysis frequencies and the transmittivities of $J$ values in the reference (Ultem coated) and reaction (clear) chambers, respectively. The "transmittivities" column shows the transmittivities of the tested species from the measurements conducted with the set photolysis frequencies using SERIC XG-500B sunlight (this study) and ambient (literature). It should be noted that the errors listed here are relatively large and may not reliable due to a limit number of measurement points (3 points for each species). The calculated transmittivity errors are 0.07 for all species

Furthermore, $P(O_3)$net error caused by the ±5 % measurement error of $J$ values during the 3rd stage of the 4-min simulation in method I are discussed in Sect. 3. in the modified manuscript on page 30, lines 682-689:

"To evaluate $P(O_3)_{net}$ error caused by the measurement error of $J$ values, we introduced a ±5 % error to the measured $J$ values during the 3rd stage of the 4-min simulation in method I. The modeled $P(O_3)_{net}$ results are presented in Fig. S24 in the supplementary materials. We observed that the inclusion of a -5 % measurement error in $J$ values led to a decrease in $P(O_3)_{net}$ by 7.27 %, while adding a +5 % measurement error in $J$ values caused an increase in $P(O_3)_{net}$ by 3.08 %. This implies that the maximum bias of $P(O_3)_{net}$ caused by the measurement error of $J$ values falls within the error range of the currently assessed $P(O_3)_{net}$ error, which was 13.9 % for method I. Therefore, we conclude that this type of error will not influence our final modeling results and conclusions."

And the modeled $P(O_3)_{net}$ results are added in Fig. S24 in the modified supplementary materials on page 31, lines 395-399:

"

[Figure]

**Figure S24: $P(O_3)_{net}$ changing in the reaction and reference chambers in method I with ± 5 % of measured $J$ values.**"

Appendix:

We detected an error in the Supplementary Materials Table S12. The equation of measured $J$ values used in the model simulation for Method II was incorrectly written, which we have now corrected. It is important to note that this didn't

influence any of our analyses results as we used the correct equation throughout all our analyses.

**Table S12. *J* values used in the model simulation in reaction and reference chambers.**

| | *J* values used in the model simulation | |
|---|---|---|
| | Measured *J* values: $J(NO_2)$, $J(O^1D)$, $J(HONO)$, $J(H_2O_2)$, $J(NO_3\_M)$, $J(NO_3\_R)$, $J(HCHO\_M)$, $J(HCHO\_R)$ | Unmeasured *J* values: $J(HNO_3)$, $J(CH_3CHO)$, $J(MACR)$, $J(MEK)$, $J(HOCH_2CHO)$, $J(C_2H_5CHO)$, $J(C_3H_7CHO)$, $J(C_4H_9CHO)$, etc. |
| Method I | $J_{trans\ measured} \times J_{value\ measured}$ | $J_{trans\ TUV} \times J_{NO2\ measured} / J_{NO2_{TUV}} \times J_{valueTUV}$ |
| Method II | $J_{trans\ TUV} \times J_{value\ measured}$ | $J_{trans\ TUV} \times J_{NO2\ measured} / J_{NO2_{TUV}} \times J_{valueTUV}$ |

"

References

Bohn B, Lohse I. Calibration and evaluation of CCD spectroradiometers for ground-based and airborne measurements of spectral actinic flux densities. Atmos. Meas. Tech., 10, 3151–3174, https://doi.org/10.5194/amt-10-3151-2017, 2017.

Burkholder, J., Sander, S., Abbatt, J., Barker, J., Cappa, C., Crounse, J., Dibble, T., Huie, R., Kolb, C., and Kurylo, M.: Chemical kinetics and photochemical data for use in atmospheric studies, evaluation number 19, Pasadena, CA: Jet Propulsion Laboratory, National Aeronautics and Space Administration, 2020, http://jpldataeval.jpl.nasa.gov. 2020.